



# Retrieval of aerosol optical properties over Beijing: a comparison between SKYNET and AERONET

Xianyi Yang[1,2], Huizheng Che[2], Hitoshi Irie[3], Quanliang Chen[1], Ke Gui[2], Ying Cai[3], Yu Zheng[2], Linchang An[4], Hujia Zhao[2], Lei Li[2], Yuanxin Liang[2], Yaqiang Wang[2], Hong Wang[2], Xiaoye Zhang[2]

[1]Plateau Atmospheric and Environment Laboratory of Sichuan Province, College of Atmospheric Science, Chengdu University of Information Technology, Chengdu 610225, China
[2]State Key Laboratory of Severe Weather (LASW), Institute of Atmospheric Composition, Chinese Academy of Meteorological Sciences, Beijing 100081, China
[3]Center for Environmental Remote Sensing, Chiba University, Chiba, Japan
[4]National Meteorological Center, CMA, Beijing 100081, China

*Correspondence to*: Huizheng Che (chehz@cma.gov.cn)

**Abstract.** This study assesses the performance of SKYNET in comparison to AERONET (Aerosol Robotic Network) for retrieving aerosol optical properties (AOPs) in Beijing, China. The results obtained from simultaneous measurements show high correlation coefficients (> 0.994) for aerosol optical depth (AOD) at each wavelength. The highest correlation

coefficient for Ångström exponent is 0.825, at 500–870 nm. The single scattering albedo (SSA) of SKYNET is systematically larger than that of AERONET at each wavelength, and adjusting the SVA (solid view angle) and SA (surface albedo) input values can easily affect the value of SKYNET SSA. The volume size distribution patterns derived from the two networks' instruments are both bimodal, which is typical, while the coarse-mode volume of SKYNET is larger than that of AERONET on average.

According to the frequency distribution of aerosol particles, coarser aerosol particles often present in autumn and finer particles usually exist in winter, and there are more absorbent aerosol particles in winter. SKYNET data, combined with meteorological data, CALIPSO (Cloud–Aerosol Lidar and Infrared Pathfinder Satellite Observations) data, backward trajectories, and WPSCF (weighted potential source contribution function) and WCWT (weighted concentrated weighted trajectory) analyses are used to analyze a serious pollution event in winter over Beijing. The results suggest that it was not

only affected by local emissions but also by regional transport. The AOPs under three weather conditions (clean, dusty, haze) in Beijing are discussed. The values of AOD on haze days are about 10.3, 10.0, 8.7, 6.3 and 6.2 times larger than those on clean days at 400, 500, 670, 870 and 1020 nm, respectively; and under haze conditions, the $PM_{2.5}$ (fine particulate matter) is about 7.6 times larger than that under clean conditions. The values of AOD on dusty days are about 7.1, 7.4, 7.0, 5.3 and 5.2 times larger than those on clean days at 400, 500, 670, 870 and 1020 nm, respectively; and under haze conditions, the $PM_{2.5}$

is about 5.2 times larger than that under clean conditions.



## 1 Introduction

Aerosol particles have an important influence on the Earth's radiation budget because they can scatter and absorb solar radiation (Sokolik et al., 2001; Ackerman and Toon, 1982) and affect the microphysical processes of clouds (Hansen et al., 1997; Charlson et al., 1992; Li et al., 2011). It has been reported that aerosol particles not only have an impact on global and

regional climate change (Hansen et al., 2000), but also cause polluted environments and adversely affect human health (Uchiyama et al., 2018; Sun et al., 2018). AOD has become a key indicator for assessing global climate change and the variation in meteorological parameters can determine the inter-decadal change in regional AOD (Che et al., 2019). Despite many studies on aerosols, their concentrations and optical properties and large variability in time and space (Uchiyama et al., 2018) are collectively one of the largest sources of uncertainty in current assessments and predictions of global climatic

change (Stocker et al., 2001; Hansen et al., 2000; Ramanathan et al., 2001; Che et al., 2008).

Ground-based measurement networks are a very useful and accurate way to monitor the spatiotemporal distribution of aerosols (Holben et al., 2001) by using the sun-sky radiometric technique (Holben et al., 1998). SKYNET (Nakajima et al., 2007; Takamura and Nakajima, 2004), located mostly in Asia and Europe, is an observation network dedicated to aerosol–cloud–radiation interaction research (Nakajima et al., 1996; Nakajima et al., 2007; Che et al., 2014). AERONET (Aerosol

Robotic Network; Holben et al., 2001) is another well-known ground-based remote-sensing aerosol network, established by NASA (National Aeronautics and Space Administration) and PHOTONS (Photométrie pour le Traitement Opérationnel de Normalisation Satellitaire). SKYNET and AERONET measure direct and diffuse solar radiation by using skyradiometers (Prede Co., Ltd, http://prede.com/english/index.htm) and sunphotometers (CE-318, Cimel, https://www.cimel.fr/), respectively (Holben et al., 1998; Uchiyama et al., 2005) to retrieve aerosol optical properties (AOPs; Kim, 2004; Nakajima,

2003; Takemura et al., 2002; Dubovik et al., 2002; Eck et al., 2005; Holben et al., 2001; N. T. et al., 2000; Smirnov et al., 2002). The radiative and optical properties of aerosols in the atmospheric column can be retrieved from measurements including the direct irradiance from the sun and the diffuse radiance from the sky by the sky-sun radiometric technique (Estellés et al., 2010). AOPs retrieved from skyradiometer measurements have been used to compare with other ground-based measurements, such as MAX-DOAS (Multi-Axis Differential Optical Absorption Spectroscopy) measurements (Irie et

al. 2008, 2015), autonomous marine hyperspectral radiometers (Wood et al., 2017), and satellite products, including those from Himawari-8 (Yumimoto et al., 2016; Damiani et al., 2018). The aerosol optical depth (AOD) can be inferred from the direct sun measurements, and other AOPs, including single scattering albedo (SSA), the real and imaginary parts of the refractive index, the volumes of different aerosol particle size distributions, and the scattering phase function, can be retrieved from a combination of direct sun and diffuse sky radiance measurements (Che et al., 2014). Pandithurai et al. (2009)

retrieved the cloud effective radius and optical thickness from skyradiometer measurements.

The measurement protocols and retrieval algorithms of SKYNET and AERONET differ from one another, so it is important to make sure AOPs are consistent between the two networks (Che et al., 2008). Sano et al. (2003) found that the difference in AOD was less than those of other AOPs, and that an effective comparison between the two networks'



instruments needs at least one year of measurements. Campanelli et al. (2004) compared the AOPs between the two networks' instruments, but the results for refractive index were not good because of the few measurements and immature algorithm. Evgenieva et al. (2008) compared the AOPs between AERONET and SKYNET using two days of measurements and found the lowest deviation to be at 675 nm. Estellés et al. (2010) compared the AOD between AERONET and two modes of ESR (EuroSkyRad) and concluded that such a comparison must have a long database. Estellés et al. (2012) then compared the differences between AERONET and SKYRAD4.2 inversion products retrieved from one month of Cimel data.

With the rapid development of both science and the economy over the past three decades in China, the processes of industrialization and urbanization have accelerated, and human activities are becoming more frequent. As a result, serious environmental problems such as haze and dust are found everywhere, but especially in Beijing—the capital of China. Numerous studies have investigated haze and dust events in Beijing based on the optical properties of aerosol. However, these studies have mainly focused on the AOPs retrieved from sunphotometers (i.e. from AERONET), with few having analyzed the AOPs retrieved from skyradiometers (i.e. SKYNET).

The aim of this work is to compare and assess simultaneous observations of AERONET sunphotometers and SKYNET skyradiometers in Beijing. Then, we analyze a serious pollution event that occurred in the Beijing region from 27 December 2016 to 9 January 2017 based on the SKYNET skyradiometer measurements and using the skyradiometer analysis package from the Center for Environment Remote Sensing (SR-CEReS), version 1 (Mok et al., 2018).

## 2 Data and methods

### 2.1 Instrumentation and Protocols

A skyradiometer (POM-02, Prede Co. Ltd, http://prede.com/english/index.htm) and a sunphotometer (CE-318, Cimel, https://www.cimel.fr/) were installed at the Chinese Academy of Meteorological Sciences (116.317 °E, 39.933 °N) in Beijing in September 2016 and March 2007 as part of SKYNET and AERONET, respectively, to measure AOPs, and have been working continuously since then. This site lies in the northwest of Beijing, which is a region filled with human activities (Fig. 1). The skyradiometer is an automatic instrument measuring the direct solar irradiance and the diffuse sky radiance with a 1.0 ° full field-of-view at eleven channels (315, 340, 380, 400, 500, 670, 870, 940, 1020, 1600, and 2200 nm). It was programmed to acquire direct sun irradiance every minute and sky diffuse radiance every 10–20 minutes (Estellés et al., 2010). The sunphotometer is another automatic instrument for tracking the sun and scanning the sky, but with a 1.2 ° full field-of-view at four observation channels (440, 670, 870, and 1020 nm), three 870-nm polarization channels, and a water vapor channel (940 nm) (Holben et al., 1998). In this study, SKYNET data from five channels (400, 500, 670, 870, and 1020 nm) are used to retrieve AOPs.



## 2.2 Retrieval method

In this study we use the SR-CEReS analysis package, which is released via the SKYNET website (http://atmos3.cr.chiba-
u.jp/skynet/index.html). As the main program of the package, SKYRAD.pack, version 5.0 (Hashimoto et al., 2012), is used
to retrieve the AOPs. SKYRAD.pack algorithm was originally implemented (version 4.2) by Nakajima et al. (1996), and
subsequently modified (version 5.0) by Hashimoto et al. (2012). These SKYRAD.pack algorithms assume aerosols are
spherical particles. On the other hand, the algorithm developed by Dubovik and King, (2000) and Dubovik et al. (2000, 2006)
is used by AERONET to retrieve AOPs. The AERONET algorithm accounts for aerosol non-sphericity by applying a
spheroid model (Dubovik et al., 2006).

An improved version of the Langley plot (referred to as the "ILP") is used to calibrate the skyradiometer calibration
constant $F_0$ *in situ*. Campanelli et al. (2004) presented a new procedure for the *in situ* determination of the solar calibration
constant, and the precision of the method—by testing a five-month dataset obtained from a Prede skyradiometer in Rome,
Italy—was estimated to fall within 1%–2.5%. The SR-CEReS software developed by Chiba University selects the input data
for the ILP method more carefully than before, and SKYRAD.pack version 5.0 is an open source software package released
on the OpenCLASTR website (http://157.82.240.167/~clastr/), so we opt to use the two pieces of software to retrieve the
measurements from the SKYNET skyradiometer respectively, allowing us to compare the degree of correlation between the
two SKYNET-retrieved AOD and AERONET-retrieved AOD respectively.

An efficient radiative transfer code, as well as a linear and nonlinear inversion scheme, can be used to derive the AOD,
SSA, complex refractive index, and size distribution (Nakajima et al., 1996). The AOD, $\tau_{a(\lambda)}$, is defined as

$$\tau_{a(\lambda)} = \int_{r_m}^{r_M} \pi r^2 \, Q_{ext}(x, \widetilde{m}) n(r) dr, \tag{1}$$

where $Q_{ext}$ is the efficiency factor for extinction, as given by Mie theory for spherical particles; $x = 2\pi r/\lambda$ is the size
parameter; $n(r)$ is the columnar radius distribution of aerosol; $r_m$ and $r_M$ are the minimum and maximum aerosol radii,
respectively; and $\widetilde{m} = m - k$i is the aerosol complex refractive index, wherein $m = 1.50$ and $k = 0.005$ are used in the
code. The extinction includes scattering and absorption. If $Q_{scatter}$ takes the place of the efficiency factor in Eq. (1), the
optical depth for scattering ($\tau_{as(\lambda)}$) can be calculated and the aerosol SSA defined as $\omega_a = \tau_{as(\lambda)}/\tau_{a(\lambda)}$ (Nakajima et al.,
1996). The volume particle size distribution for SKYNET and AERONET is derived in 20 and 22 logarithmically equidistant
bins in the size ranges of $0.012\mu m \leq r \leq 16.540\mu m$ and $0.05\mu m \leq r \leq 15\mu m$, respectively. The columnar volume
spectrum is defined as

$$\frac{dV}{dlnr} = \frac{V_0}{\sigma\sqrt{2\pi}} exp\left[-\frac{(ln(r/r_m))^2}{2\sigma^2}\right], \tag{2}$$

where σ, r and $r_m$ are the standard deviation of particles, aerosol radius, and volume median radius, respectively (Dubovik et
al., 2002; Kim, 2004). The Ångström exponent can be defined by Ångström's law as $\tau = \beta\lambda^{-\alpha}$, where $\alpha$ is the Ångström
exponent, which is an index of size distribution. A high Ångström exponent indicates that the relative number of small
particles is more than that of large particles in the size distribution, and vice versa (Uchiyama et al., 2005).


Owing to the fact that the SKYNET skyradiometer lacks the AOD at the 440-nm wavelength, we calculate it by interpolating the AOD at 400 nm and 500 nm. The formulae for doing so are as follows:

$$\alpha = -\frac{lg\left(\frac{\tau_{a(400)}}{\tau_{a(500)}}\right)}{lg\left(\frac{400}{500}\right)}, \tag{3}$$

$$\tau_{a(440)} = \tau_{a(400)} \times \left(\frac{440}{400}\right)^{-a}, \tag{4}$$

where $\alpha$ is the Ångström exponent and $\tau_{a(440)}$ means the AOD at 440 nm.

## 2.3 Meteorological data


Gui et al. (2019) found the highest regional mean $PM_{2.5}$ (fine particulate matter) concentrations for the period 1998–2016 to have occurred over the North China Plain. To monitor the PM mass concentration, an ambient particulate monitor (TEOM 1405, ThermoFisher, https://www.thermofisher.com) that acquired hourly PM data for the study region in Beijing during the analysis period was used, and the data were obtained from the China Environmental Monitoring Center.

Reanalysis data for the wind fields (ERA-Interim) were downloaded from the European Centre for Medium-Range Weather Forecasts (https://apps.ecmwf.int/datasets/). The daily wind field at the surface and 850 hPa were used to analyze the region with a spatial resolution of $0.125° \times 0.125°$.

The 72-hour $PM_{2.5}$ backward trajectories at multiple altitudes over Beijing were calculated using archived data from the FNL    (final)    analyses    of    the    NCEP    (National    Centers    for    Environmental    Prediction;

ftp://arlftp.arlhq.noaa.gov/pub/archives/gdas1/)    and    the    TrajStat    software (http://www.meteothink.org/downloads/index.html), which applies cluster methods to airmass backward trajectories (Sirois and Bottenheim, 1995) to infer the major transportation pathways. Furthermore, the WPSCF (weighted potential source contribution function) and WCWT (weighted concentration weighted trajectory) models were used to add the PM concentrations into the backward trajectories to analyze the pollutant sources and main transportation pathways. Wang et al.

(2006) describe these two models in detail.

## 2.4 Satellite data

The CALIOP (Cloud–Aerosol Lidar with Orthogonal Polarization) instrument installed on the CALIPSO (Cloud–Aerosol Lidar and Infrared Pathfinder Satellite Observations) satellite can provide vertical profiles of AOPs (Liu et al., 2008; Omar et al., 2009). The products of CALIPSO used in this study can be downloaded from NASA's Langley Research Center website

(https://www-calipso.larc.nasa.gov/). The overall characteristics of atmospheric aerosols can be determined by the vertical profiles of aerosol-type data, so this product can distinguish the aerosol types. Aerosol types such as dust, pollution dust, smoke, marine etc. are defined in the CALIPSO models. For this study, we selected the version 4.10 CALIPSO products to depict aerosol subtypes.



## 3. Results and analysis

**3.1 Comparison of AOP retrievals between SKYNET and AERONET**

The level-2.0 AOD retrieved by the version-3 direct sun and inversion algorithm from AERONET is used for comparison with the AOD retrieved by SKYNET. To ensure a proper comparison, only those data observed by the networks' two instruments within one minute of each other should be used, which yields 561 measurements in total. Figure 2 shows the results of two inversions of SKYNET compared with AERONET. Figures 2a–d show that the AOD retrieved from the

AERONET sunphotometer, at all wavelengths, is systematically lower than that retrieved from the SKYNET skyradiometer, and the MBD (mean bias deviation, defined as $mean_{AERONET} - mean_{SKYNET}$) at 500 nm, 670 nm, 870 nm and 1020 nm is $-1.7\%$, $-1.7\%$, $-2.4\%$ and $-3.0\%$, respectively. However, Figures 2e–h illustrate that the AOD retrieved from the AERONET sunphotometer, at all wavelengths, is systematically higher than that retrieved from the SKYNET skyradiometer, and the MBD at 500 nm, 670 nm, 870 nm and 1020 nm is $6.4\%$, $3.3\%$, $1.8\%$ and $1.5\%$, respectively. The correlation of AOD

has been obviously improved, by $0.91\%$, $1.32\%$, $2.05\%$ and $1.64\%$ at 500 nm, 670 nm, 870 nm and 1020 nm, respectively. The correlation coefficient of AOD between the SKYNET SR-CEReS retrieval and AERONET at each channel is larger than 0.994. This significant coefficient confirms that the two networks' instruments are highly consistent in their measurement of AOD. Additionally, selecting the input data for the ILP method more carefully can improve the data quality and the correlation of AOD between the SKYNET skyradiometer and AERONET sunphotometer.

As shown in Fig. 3, the correlation coefficient of Ångström exponent at 440–670 nm ($\alpha_{440-670nm}$) is 0.720. Also, the relatively high correlation coefficient of Ångström exponent between the networks' two instruments, at $\alpha_{440-870nm}$ and $\alpha_{500-870nm}$, is 0.821 and 0.825, respectively. The correlation coefficients of $\alpha_{440-670nm}$ and $\alpha_{440-870nm}$ are both lower than that of $\alpha_{500-870nm}$, probably because the SKYNET skyradiometer does not have an AOD value at 440 nm, which is instead calculated by interpolation. The Ångström exponent of AERONET is larger than that of SKYNET.

The level-1.5 data retrieved by the version-3 direct sun and inversion algorithm from AERONET are used to compare with the SSA, complex refractive index, and volume size distribution retrieved by SKYNET. Owing to the fact that the AERONET sunphotometer measures the daily sky radiance less frequently than the SKYNET skyradiometer, the observation interval is considered to be simultaneous within 3 minutes. A comparison of the SSA results between the networks' two instruments is shown in Fig. 4. Importantly, the SSA from SKYNET at both 400 nm and 500 nm correlate to

the SSA at 440 nm from AERONET. The correlation coefficients are 0.502, 0.665, 0.606, 0.565 and 0.614 at 400, 500, 670, 870 and 1020nm, respectively. Although the correlation of SSA at each channel is not as good as that of AOD, there are still linear relationships that can be seen from the results. The MBD of SSA at each channel is about $-2.7\%$, $-5.7\%$, $-4.0\%$, $-3.1\%$ and $-3.5\%$, which means the SSA from the SKYNET skyradiometer is larger than that from the AERONET sunphotometer.

Through the above comparisons we can see that the AOD retrieved from AERONET and SKYNET shows significant

correlation and a high degree of consistency, but the SSA and complex refractive index retrieved from the two algorithms bear some differences. The factors affecting the inconsistent SSA between SKYNET and AERONET were studied by Khatri



et al. (2016), who stated that underestimating the calibration constant for sky radiance is the most likely reason for the high SSA at each channel in SKYNET, and SA is regarded as a secondary effect of SSA (Khatri et al., 2016). In this study, several sensitivity tests were carried out to test the magnitude of the change in SSA (see Supplement). The solid view angle

(SVA) calculated by the SR-CEReS analysis package was considered as the experimental group, while values of SVA sequentially changed by 1% were regarded as the control group. One can see that, at the same moment, the smaller SVA values lead to larger SSA values (Fig. S1). The SVA is related to the sky radiance, and errors in the SVA will affect the SSA results. Overestimation or underestimation in the SVA results in underestimation or overestimation of the SSA, respectively (Uchiyama et al., 2018). With the SVA reduced by 4%, the SSAs are about 0.0445 (5.96%), 0.0287 (3.38%) and 0.0175

(1.88%) larger than those from experimental group at 400, 500 and 670 nm, respectively. On the other hand, with the SVA increased by 3%, the SSAs are about 0.0839 (9.68%), 0.0668 (7.28%) and 0.0461 (4.99%) smaller than those from the experimental group at 400, 500 and 670 nm, respectively. As shown in Fig. S2, adjusting the surface albedo (SA) also leads to the SSA changing. The SR-CEReS analysis package by default uses SA values of 0.05, 0.10 and 0.10 at 400, 500 and 670 nm, respectively, which was the experimental group, while values of SA sequentially changed by 0.01 were regarded as the

control group. It is clear that, at the same moment, the smaller SA values cause larger SSA values. Although generally the reflected light is not as important as the incident light for downward radiation, overestimation or underestimation of SA would lead to underestimation or overestimation of the SSA, respectively (Dubovik et al., 2000; Khatri et al., 2012). When the SA is reduced by 0.04 compared to the initial value, the SSAs are about 0.0263 (3.05%), 0.0285 (3.17%) and 0.0295 (3.26%) larger than those from the experimental group at 400, 500 and 670 nm, respectively. However, when the SA is

increased by 0.03, the SSAs are about 0.0138 (1.60%), 0.0183 (2.03%) and 0.0207 (2.29%) smaller than those from the experimental group at 400, 500 and 670 nm, respectively. To make SKYNET data more accurate and reliable, strict control over a suitable SA and accurate SVA value is necessary. It is critical, therefore, for a suitable method to be found for SKYNET that determines these values.

A comparison of the imaginary part of the complex refractive index ($m_i$) retrieved from SKYNET and AERONET is

shown in Fig. 5 (note that $m_{is400}$ and $m_{is500}$ from SKYNET correlate to $m_{is440}$ from AERONET). The correlation coefficients between SKYNET and AERONET at all wavelengths are 0.565, 0.565, 0.463, 0.459 and 0.611, respectively. In contrast to the SSA results, the values of the imaginary part of the complex refractive index derived from the SKYNET skyradiometer at each channel are lower than those from the AERONET sunphotometer. The mean values of the imaginary part of the complex refractive index retrieved from AERONET are about 0.005, 0.008, 0.004, 0.003 and 0.003 larger than

those from SKYNET at 400, 500, 670, 870 and 1020 nm, respectively. The absorption capacity of aerosol particles can be expressed by the imaginary part of the complex refractive index. It can be inferred that the SKYNET-derived aerosol particles would have less absorption than the AERONET ones.

An intercomparison of the real part of the complex refractive index ($m_r$) between SKYNET and AERONET is shown in Fig. 6. The correlation coefficients are 0.410, 0.557, 0.581, 0.677 and 0.582 for $m_{rs400}$ with $m_{ra440}$, $m_{rs500}$ with $m_{ra440}$,

$m_{rs670}$ with $m_{ra670}$, $m_{rs870}$ with $m_{ra870}$, and $m_{rs1020}$ with $m_{ra1020}$, respectively, where $m_{rs}$ and $m_{ra}$ represent the real part





of the refractive index of SKYNET and AERONET. Clear linear correlations between SKYNET and AERONET can be seen at all channels in Fig. 6. The mean values of the real part of the complex refractive index retrieved from AERONET are about 0.036, 0.034, 0.022, 0.005 and 0.010 lower than those from SKYNET at 400, 500, 670, 870 and 1020 nm, respectively. The highest correlation and the smallest MBD value of the real part of the complex refractive index are at channel of 870 nm.

A comparison of the volume size distribution between AERONET and SKYNET is shown in Fig. 7, wherein only those data observed within 3 minutes of each other were considered as simultaneous, which gave the number of eligible data during the whole observation period as 313. The volume of aerosol for an air column of unit cross section is used to express the columnar volume spectrum ($dV/dlnr$), and the radius is in logarithmic form (Nakajima et al., 1996). There are differences in the assumptions of size distribution between the SKYNET and AERONET retrieval algorithms. The volume at

each rated radius is calculated by averaging the values at that radius for both the SKYNET skyradiometer and the AERONET sunphotometer. However, the number of rated radii for SKYNET and AERONET is 20 and 22, respectively, meaning 20 rated radii (0.012, 0.018, 0.026, 0.038, 0.055, 0.081, 0.118, 0.173, 0.253, 0.370, 0.541, 0.791, 1.156, 1.691, 2.473, 3.617, 5.289, 7.734, 11.310 and 16.540 μm) are used to retrieve the volume size distribution for SKYNET and 22 rated radii (0.050, 0.066, 0.086, 0.113, 0.148, 0.194, 0.255, 0.335, 0.439, 0.576, 0.756, 0.992, 1.301, 1.708, 2.241, 2.940,

3.857, 5.061, 6.641, 8.713, 11.432 and 15.000μm) are used to retrieve the volume size distribution for AERONET. As is shown in Fig. 7, the size distribution patterns from SKYNET and AERONET are both bimodal, which is typical, but the peak volumes bear some differences. Specifically, the two peak volumes from the SKYNET skyradiometer are at the radii of 0.173 μm and 5.289 μm, with columnar volume spectra of 0.029 and 0.064 $\mu m^3/\mu m^2$; whereas, those from the AERONET sunphotometer are at radii of 0.148 μm and 3.857μm, with columnar volume spectra of 0.032 and 0.051 $\mu m^3/\mu m^2$. One can

see that the coarse-mode volume of SKYNET is larger than that of AERONET on average; whereas, in contrast, the fine-mode volume of SKYNET is smaller than that of AERONET on average. The SSA is a ratio that describes the scattering ability of aerosol particles and, generally, coarse-mode particles have a larger scattering ability, meaning the SSA will be larger when there are many coarse-mode particles. The difference in volume size distribution between SKYNET and AERONET might be one reason why the SSA retrieved from SKYNET is larger than that retrieved from AERONET. In

addition, the columnar volume spectrum with radius from 0.086 to 0.756 μm retrieved from AERONET is larger than that of SKYNET, meaning the volume of fine-mode particles retrieved from AERONET is larger. As can be seen from Fig. 5, the imaginary part of the complex refractive index from AERONET is larger than that of SKYNET. The larger volume of fine-mode particles of AERONET will result in greater absorption, which is probably the reason behind the larger imaginary part for AERONET than SKYNET. It can be clearly seen that the deviations of the columnar volume spectrum around the peak

volumes are larger than for other volumes, which is the same for both the SKYNET skyradiometer and the AERONET sunphotometer. The largest deviations between SKYNET and AERONET are 0.059 at the radius of 5.289 μm and 0.037 at the radius of 3.857 μm. However, the deviations for the volume of fine-mode particles retrieved from AERONET are larger than those of SKYNET in most cases; whereas, for the volume of coarse-mode particles, the deviations are larger for SKYNET than AERONET. From Fig. 7 we can see that the columnar volume spectrum retrieved from AERONET is nearly

0 $\mu m^3/\mu m^2$ at the radii less than 0.050 μm and more than 15.000 μm; however, the values retrieved from SKYNET are about 0.008 and 0.004 $\mu m^3/\mu m^2$ at these radii, respectively. In Beijing, there might be some particles whose radii are larger than the maximum values of rated radii for the SKYNET algorithm, but the AERONET algorithm ignores these particles.

### 3.2 Frequency distribution of SKYNET-retrieved AOPs in autumn and winter in Beijing

The frequency distribution of AOD values at 400 nm in autumn and winter is shown in Fig. 8a. The bin interval for the AOD
is 0.1. From the frequency histograms of AOD values at 400 nm we can see a typical bimodal pattern with two peak ranges of AOD at 0.1–0.2 and 0.6–0.7 in SON (September–October–November) and a single pattern with a peak range of AOD at 0.0–0.1 in DJF (December–January–February). The frequency distributions of AOD values between 0.0 to 0.5 are about 55.47% and 76.70% of all values in autumn and winter, respectively. During the study period, we can see that relatively high AOD values often occur in autumn rather than winter. The mean values of AOD are about 0.476 and 0.321 in autumn and
winter, respectively.

The frequency distribution of Ångström exponent in autumn and winter is shown in Fig. 8b. The bin interval is 0.1. The frequency distributions of Ångström exponent values between −0.20 and 0.80 account for about 18.37% and 16.19% of all values in autumn and winter, respectively. Meanwhile, the frequency distributions of Ångström exponent values larger than 1.4 are about 12.67% and 20.68% of all values in autumn and winter, respectively. It can be clearly seen that larger aerosol
particles often present in autumn and finer particles usually exist in winter. The mean values of Ångström exponent are 0.966 and 1.024 in autumn and winter, respectively.

The SSA frequency histogram in Fig. 8c shows various ranges of SSA for the two seasons. The bin interval for the SSA is 0.01. The frequency distributions of SSA values larger than 0.95 account for 37.19% and 28.99% of all values in autumn and winter, respectively, suggesting that there are more scattering aerosol particles in autumn.
The volume size distribution of aerosol plays a major role in observations and simulations of radiative forcing (Dusek et al., 2006). From Fig. 8d, we can see that the volume size distributions for autumn and winter show a typical bimodal pattern, for which the two peak volumes are at radii of 0.173 and 5.289, with columnar volume spectra of 0.031 (0.021) and 0.081 (0.054) for autumn (winter). Moreover, the radius corresponding to the peak of the fine-mode volume size distribution of autumn is a little larger than that of winter.
According to the frequency distributions of AOPs over Beijing for the study period, we can see that the average AOD value for the whole of autumn is larger than that for the whole of winter. In autumn, there are still dust events in Beijing (Sheng et al., 2019), and dust events can lead to high AOD values (Che et al., 2009). The frequent cold air with strong winds may lead to the lower AOD values in winter because they can accelerate the diffusion of pollutants (Giavis et al., 2005). Ångström exponent values are higher in winter, and the results indicate that the size of aerosol particles are smaller in winter.
Vertical air turbulence in autumn is stronger than that in winter, and a greater number of particles such as floating dust are transported into the upper air. Hygroscopic growth of fine-mode particles can be seen in autumn from Fig. 8d, due to more precipitation in autumn than in winter. Aerosol particles combine with water vapor to become larger and ultimately improve



the aerosol scattering ability (Yan et al., 2009). Dust events and hygroscopic growth of aerosol particles may lead to more coarse-mode particles in autumn, which can also be confirmed in the volume size distribution.

## 3.3 Combined analysis of a heavy pollution episode over Beijing

### 3.3.1 Variation in PM mass concentration and meteorological data

Figure 9a shows the temporal variation of PM mass concentration from 27 December 2016 to 9 January 2017, a period representative of winter in Beijing. The Chinese National Secondary Standards for $PM_{2.5}$ and $PM_{10}$ (coarse PM) are 24-hour average concentrations less than 75 and 150 $\mu g/m^3$, respectively. In order to clearly see the pollution processes, the figure is shaded based on this standard for $PM_{2.5}$. Before every pollution processes, explosive growth in the PM values is clear. During the non-pollution periods, the ratios change greatly, while those in the pollution periods change little. On 2 January 2017, the minimum ratio value is about 0.60, whereas that on 4 January 2017 is approximately 0.87. The diurnal variation of temperature and relative humidity (RH) shows a typical negative correlation in Fig. 9b. The RH during pollution periods is increased by more than 50% compared to in non-pollution periods. Such a condition (high RH) is conducive to the formation of secondary aerosol species such as $NO_3^-$, $SO_4^{2-}$ and secondary organic compounds (Hennigan et al., 2008; Blando and Turpin, 2000). The correlation coefficient between visibility and RH is about −0.80, showing a significant negation correlation during the study period. Visibility is less than 3 km, while RH is as high as 98.49% in the pollution periods (Fig. 9b). RH plays an important role in the hygroscopic growth and scattering ability of aerosol particles (Fu et al., 2014), which can affect visibility (Gui et al., 2016). Visibility is also influenced by meteorological conditions, such as wind direction and wind speed (Fu et al., 2014).

Figure S3 (see Supplement) shows the 850-hPa and surface daily averaged wind fields around the Beijing area in the study period. From 27 December 2016 to 3 January 2017, the direction of wind at 850 hPa is mainly northwest. The wind speed at the surface is less than 2 m s$^{-1}$, and less than 6 m s$^{-1}$ at 850 hPa, in the Beijing area on 27 December, which would have been conducive to the accumulation of pollutants. However, on 28 December, the direction of wind at the surface and at 850 hPa is in both cases northwest, and the wind speed becomes stronger (>12 m s$^{-1}$ at 850 hPa, >4 m s$^{-1}$ at the surface), accelerating the spread of pollutants and leading to the reduction in PM values and the recovery of visibility in the second half of 28 December. The strong wind (>8 m s$^{-1}$ at 850 hPa, >3 m s$^{-1}$ at the surface) blows through Beijing area during the development stage of the pollution process on 29 December, which facilitates the long-distance transmission of pollutants. The speed of wind at 850 hPa on 30 December is less than that in the pollution period (from 30 December 2016 to 3 January 2017), while $PM_{2.5}$ and $PM_{10}$ reach 442 $\mu g/m^3$ and 538 $\mu g/m^3$, respectively. The indication here is that wind speed is an important factor to the mass concentration of PM. Comparing 2 January 2017 to 4 January 2017, the direction of wind changes from northwest to southwest at 850 hPa, and from a prevailing westerly to a prevailing easterly at the surface, suggesting that the transmission route of pollutants in the Beijing area has changed. In addition, the speed of wind at 850 hPa and the surface gradually decrease, culminating in calm wind conditions on 4 January 2017. This would have been





conducive to the accumulation of pollutants and, indeed, the value of $PM_{10}$ reaches 813 μg/$m^3$. On 5 January 2017, the wind

speed is still very low, and there is a clear anticyclone at the surface around the Beijing area on 6 January 2017, which would

have limited the spread of pollutants. Thereafter, the wind direction turns northerly and the speed of the wind increases day

by day, conducive to the spread of pollutants, resulting in the reduced PM values.

### 3.3.2 Variation in AOPs

Figure 10a shows the daily averaged AOD variations from the SKYNET skyradiometer measurements and the daily

averaged $PM_{2.5}$ during the study period. The missing values of AOD were caused by the accumulation of cloud. It can be

clearly seen that the trend of AOD at each channel is similar to that of $PM_{2.5}$, and the AOD in Beijing decreased with

wavelength. From 27 December 2016 to 29 December 2016, the AOD shows a slow growth trend, but the values are lower

than 0.2 at each channel. On 30 December 2016, the AOD increases to 0.596 and 0.537 at 400 and 500 nm, respectively.

According to the volume size distribution on that day (Fig. 10d), the coarse mode dominates, with a peak volume of 0.296

$\mu m^3/\mu m^2$ at the radius of 5.289 μm. Compared to the other days, the highest coarse-mode volume indicates that there are

dust aerosol particles over Beijing area. As can be seen from Fig. 10c, the Ångström exponent is near to 0.6 on 30 December

2016, which suggests that the proportion of coarse aerosol particles in the air is very large. Moreover, the $PM_{10}$ shows a

sharp increasing trend, and the ratio is as low as 0.72, indicating a short-lived dust event happened over Beijing. The

CALIPSO satellite can show the vertical variation of aerosol subtypes (see Supplement), which can be used to distinguish

these different subtypes, such as marine, dust, and polluted dust aerosol (Omar et al., 2009; Tao et al., 2014). Figure S4

shows that was a heavy aerosol layer under 3 km, including dust, smoke and polluted dust, near Beijing (39.93 °N, 116.32 °E),

which helps explain the much higher volume of coarse-mode particles on that day. According to the volume size distribution

during 27 December 2016 to 31 December 2016, the peak fine-mode volume increases gradually from 0.003 $\mu m^3/\mu m^2$ to

0.065 $\mu m^3/\mu m^2$, which might also be related to the development of a haze event. The SSA reflects the effectiveness of

aerosol scattering in the total extinction, which is one of the most important variables in assessing the influences of aerosols

on the radiation budget (Khatri et al., 2016). As shown in Fig. 10b, the daily averaged SSA decreased with wavelength,

showing shorter wavelengths result in higher absorption (Yu et al., 2011; Gui et al., 2016; Zheng et al., 2017). Owing to the

accumulation of absorbing aerosol particles such as black carbon aerosols, which have strong absorption abilities, the SSA

shows a downward trend. Based on the variation trend of $PM_{2.5}$ values and the meteorological data, we can see that the air

quality temporarily improves on 2 January 2017, because cold air blows through the Beijing area. However, a more serious

pollution event happened from 2 January to 4 January in 2017: the daily averaged AOD at 400 nm increases sharply from

0.30 to 0.82; the daily averaged $PM_{2.5}$ value increases from 204.12 $\mu m^3/\mu m^2$ to 313.92 $\mu m^3/\mu m^2$; and the ratio is as high

as 0.82, indicating that the proportion of $PM_{2.5}$ is higher and the fine-mode aerosol particles played a key role in the

formation of this haze event. Meanwhile, the volume size distribution shows a gradual increasing trend of fine-mode peak

volume from 0.021 $\mu m^3/\mu m^2$ to 0.051 $\mu m^3/\mu m^2$, and the Ångström exponent continuously increases to 1.03 during these

three days, indicating that the proportion of fine-mode pollutants became greater. The value of Ångström exponent is almost



larger than 0.8 in the heavy haze event, which is similar to the findings of Eck et al. (2005), Xia et al. (2007) and Zheng et al. (2017). The SSA shows a sustained downward trend at each wavelength, and the values at 400 nm vary from 0.97 to 0.91.

On 8 January 2017, the AOD at 400 nm is 0.36, and it is as low as 0.11 on 9 January 2017, indicating the end of this pollution period.

Figure 11a shows the temporal variation of AOD from the SKYNET skyradiometer measurements at five wavelengths on clean (27 December 2016), dusty (2 January 2017) and hazy (4 January 2017) days, respectively. The daily averaged AOD at 400 nm is less than 0.1 on 27 December 2016, which can be treated as the background AOD of Beijing. It should be

emphasized that only when AOD at 400 nm is greater than 0.4 on 2 January 2017 do we consider it as a dusty day (the results of the calculation of the pollution data is based on the standard of this day). The daily averaged AOD values are about 0.08±0.03, 0.07±0.02, 0.06±0.02, 0.06±0.02 and 0.05±0.02 at 400, 500, 670, 870 and 1020 nm, respectively, on the clean day. The daily averaged AOD values on the dusty day are about 0.57±0.06, 0.52±0.05, 0.42±0.04, 0.32±0.03, 0.26±0.03 at 400, 500, 670, 870 and 1020 nm, respectively. The daily averaged AOD values on the hazy day are about 0.82±0.07,

0.70±0.06, 0.52±0.05, 0.38±0.04, 0.31±0.03 at 400, 500, 670, 870 and 1020 nm, respectively. Under the three weather conditions, the AOD values show a decreasing trend with wavelength. The daily averaged AOD values on the haze day are about 10.3, 10.0, 8.7, 6.3, 6.2 times larger than those on the clean day at 400, 500, 670, 870 and 1020 nm, respectively, while the daily averaged AOD values on the dusty day are about 7.1, 7.4, 7.0, 5.3, 5.2 times larger than those on the clean day at 400, 500, 670, 870 and 1020 nm, respectively.

A comparison among the daily variations in SSA at 400 and 500 nm on clean, dusty and hazy days is depicted in Fig. 11b. The daily averaged SSA values on the clean day are about 0.996±0.003, 0.996±0.002 at 400 and 500 nm, respectively, and the values vary from 0.989 to 0.999 at 400 nm and from 0.990 to 0.999 at 500 nm, which shows a steady variation under clean weather conditions. On the dusty day, the daily averaged SSA values are about 0.94±0.02, 0.97±0.01 at 400 and 500 nm, respectively, while the values range from 0.91 to 0.99 at 400 nm, and 0.94 to 0.99 at 500 nm, indicating the SSA at

shorter wavelengths fluctuates more and is more influenced by the weather condition. The daily averaged SSA values are about 0.91±0.02 and 0.94±0.01, and the values vary from 0.89 to 0.94 and 0.93 to 0.97, at 400 and 500 nm, respectively, for the hazy day. The results show that the SSA on polluted days will fluctuate a lot more than that on clean days.

The temporal variations of Ångström exponent between 440 and 870 nm on clean, dusty and hazy days are shown in Fig. 11c. The daily averaged values of Ångström exponent are about 0.45±0.28, 0.79±0.01 and 1.03±0.02 on clean, dusty

and hazy days, respectively. The daily variation of Ångström exponent ranges from 0.04 to 0.88, 0.78 to 0.82, and 0.99 to 1.07 on clean, dusty and hazy days, respectively. The Ångström exponent on the hazy and dusty day vary smoothly, while on the clean day it varies considerably. The volume size distribution of aerosol particles retrieved from the SKYNET skyradiometer is shown in Fig. 10d. Furthermore, the volume size distributions on clean, dusty and hazy days are selected to show individually. On the clean day the volume size distribution is a typical single-modal pattern, indicating the proportion

of coarse-mode particles is much larger, which results in the Ångström exponent being very low. Both the dusty and hazy day demonstrate a classic bimodal pattern. The volume of fine-mode particles on the hazy day is larger than that on the clean





and dusty day, and the Ångström exponent on the hazy day is around 1.0, which clearly indicates fine particles have an important influence on hazy days. The Ångström exponent is as low as 0.8 on the dusty day, indicating the contribution of coarse particles.

### 3.3.3 Analysis of backward trajectory and potential pollution sources

Figure 12 shows the backward trajectories of aerosol particles and the potential pollution sources during 27 December 2016, 2 January 2017, and 4 January 2017. It is clear that there are few pollutants transported through the Beijing area on 27 December 2016 (Fig. 12a). From Fig. 12b, on 2 January 2017, cluster 1 contributes the maximum proportion of 72.50%, which originated from the desert nearby Balkhash Lake at an altitude of about 3000 m, and arrived in China from the northwestern area of Xinjiang Province, having crossed Mongolia and passed through Inner Mongolia, Hebei Province, before finally being transported eastward to Beijing. Cluster 2 contributes a proportion of 27.50%, and the airmass was originally from eastern Xinjiang Province, passing through Inner Mongolia, Hebei Province, and traveling eastwards to the Beijing area. The WPSCF result indirectly reflects the impact of local emissions on the concentration of PM$_{2.5}$ in Beijing. The high-value regions (>0.6) are mostly distributed at the eastern edge of Xinjiang Province and western Inner Mongolia, which are far away from Beijing. The WCWT method can reflect the distribution of the concentration of PM$_{2.5}$ in the transmission path to Beijing. The high-value regions (>75 μg/$m^3$) are similar to the WPSCF pattern. The dust aerosol particles come from Inner Mongolia, Xinjiang Province, and the border areas of China, playing an important role in Beijing. The analysis of backward trajectories and potential pollution sources on 2 January 2017 helps to reveal that the larger proportion of coarse particles was due to the transmission of airmass.

Trajectories during 4 January 2017 (Fig. 12c) were calculated into two clusters. Cluster 1 contributes a proportion of 60.00%, which originated from the border between Xinjiang Province and Tibet at an altitude near 3000 m, and passed through Qinghai Province, Gansu Province, Shanxi Province, and Hebei Province, traveling eastwards to the Beijing area. Cluster 2 contributes 40.00%, and the airmass was originally from Shanxi Province at a relatively low altitude, crossing Hebei Province and transporting northwards to Beijing. The high-value regions (>0.6) of WPSCF are mostly concentrated in the south of Beijing and Hebei Province, where high levels of industrialization and frequent anthropogenic activities exist, and increasing concentrations of absorptive aerosols are found owing to anthropogenic aerosol emissions (Gui et al., 2017). Similar to the WCWT pattern, the high-value areas (>75 μg/$m^3$) are also distributed over the North China Plain, which shows that short-distance transmissions from the southern area of Beijing and local emissions are the main reason for the high PM$_{2.5}$ in Beijing over this period.

### 4. Discussion

The AOPs from SKYNET retrieved by the SR-CEReS software are highly consistent with those from AERONET, apart from SSA. Sky radiance is found to be the most likely reason for the high SSA in SKYNET, with the SA regarded as the secondary reason (Khatri et al., 2016). In this study, we perform sensitivity tests by adjusting the input values of the SVA





and SA to compare the differences in the retrieved SSA, revealing that the SVA has a greater effect than the SA on SSA; the

effect of the SVA on SSA manifests as a weakening, while the effect of the SA on SSA manifests as a strengthening, along

with the increase in wavelength. A suitable method to calculate the SVA value and set an accurate SA value are needed in

future research.

Discontinuous observations and poor data quality during pollution periods cause difficulties in analyzing pollution

processes. In this study period, the SKYNET skyradiometer accurately captured a dust event, and the variation of AOPs

reflected the evolution of the pollution process effectively. Furthermore, the number of daily measurements of sky radiance

by the SKYNET skyradiometer was more than that of the AERONET sunphotometer, and thus it is an advantage for

SKYNET to use SSA values to analyze the daily variation. Therefore, accurate SSA is one of the key parameters for

analyzing pollution processes.

## 5. Summary

Based on long-term SKYNET skyradiometer and AERONET sunphotometer observations in Beijing, a comparison of AOPs

between these two networks/instruments is presented in this paper. The effect of the SVA and SA on SSA are discussed. The

frequency distributions of AOPs for SKYNET in Beijing are shown. SKYNET data, combined with meteorological data,

CALIPSO satellite data, backward trajectories, and WPSCF and WCWT analyses, are used to analyze the evolution of a

pollution process in Beijing.

Owing to the more careful selection of input data for the ILP method than before, the correlation coefficient between

the SR-CEReS-retrieved SKYNET and AERONET AOD has improved by 0.91%, 1.32%, 2.05% and 1.64% at 500 nm, 670

nm, 870 nm and 1020 nm, respectively, and is larger than 0.994 at each channel. The correlation coefficients of Ångström

exponent are about 0.720, 0.821 and 0.825 at 440–670nm, 440–870nm and 500–870nm, respectively. SSA values retrieved

by the SR-CEReS software are 2.7%, 5.7%, 4.0%, 3.1% and 3.5% larger than those from AERONET at 400, 500, 670, 870

and 1020 nm, respectively. The highest correlation coefficient of the imaginary part of the complex refractive index is 0.611,

at 1020 nm, and the values derived from the SKYNET skyradiometer at each channel are lower than those from the

AERONET sunphotometer. The highest correlation coefficient of the real part of the complex refractive index is 0.677, at

the channel of 870 nm. The volume size distribution patterns of SKYNET and AERONET are both bimodal, which is typical,

with multi-lognormal distributions. The coarse-mode volume of SKYNET is larger than that of AERONET on average;

whereas, in contrast, the fine-mode volume of SKYNET is smaller than that of AERONET on average.

Based on sensitivity tests, it was found that SSA can be easily affected. Specifically, when the SVA was reduced by 4%,

the SSAs were about 0.0445 (5.96%), 0.0287 (3.38%) and 0.0175 (1.88%) larger than those from the experimental group at

400, 500 and 670 nm, respectively. In contrast, when the SVA was increased by 3%, the SSAs were about 0.0839 (9.68%),

0.0668 (7.28%) and 0.0461 (4.99%) smaller than those from the experimental group at 400, 500 and 670 nm, respectively.

When the SA was reduced by 0.04 compared to the initial value, the SSAs were about 0.0263 (3.05%), 0.0285 (3.17%) and

0.0295 (3.26%) larger than those from the experimental group at 400, 500 and 670 nm, respectively. However, when the SA



was increased by 0.03, the SSAs were about 0.0138 (1.60%), 0.0183 (2.03%) and 0.0207 (2.29%) smaller than those from the experimental group at 400, 500 and 670 nm, respectively.

The frequency distribution of AOD values between 0.0 to 0.5 account for about 55.47% and 76.70% of all values in
autumn and winter, respectively. The frequency of Ångström exponent between −0.20 and 0.80 is larger than 18.37% in autumn, while the frequency of Ångström exponent greater than 1.4 exceeds 20.68% in winter, indicating that coarser aerosol particles often present in autumn and finer particles usually exist in winter. The frequency distributions of SSA values within the range of 0.95 to 1.00 account for 37.19% and 28.99% of all values in autumn and winter, respectively, indicating that there are more scattering aerosol particles in autumn. The volume size distributions show a typical bimodal
pattern, and the two peak volumes are at radii of 0.173 and 5.289, with columnar volume spectra of 0.031 (0.021) and 0.081 (0.054) for autumn (winter).

The haze event that occurred in Beijing from 27 December 2016 to 9 January 2017 was not only affected by local emissions, but also by regional transport. The relatively static wind at the surface facilitated the accumulation of locally emitted pollutants that were mainly absorbent aerosol particles, such as black carbon aerosol, resulting in a decreasing trend
of SSA varying from 0.97 to 0.91. According to an analysis of the backward trajectories and potential pollution sources, dust aerosol particles from the northwestern deserts were transmitted a long distance to Beijing, while absorbent aerosol particles from the North China Plain were transmitted a short distance.

It was found that the values of AOD on haze days are about 10.3, 10.0, 8.7, 6.3, 6.2 times larger than those on clean days at 400, 500, 670, 870 and 1020 nm and, under hazy conditions, the $PM_{2.5}$ is about 7.6 times larger than that under clean
conditions. The daily averaged AOD values on dusty days are about 7.1, 7.4, 7.0, 5.3, 5.2 times larger than those on clean days at 400, 500, 670, 870 and 1020 nm, respectively, and the $PM_{2.5}$ is about 5.2 times larger than that under clean conditions. The values of SSA vary from 0.89 to 0.94, 0.93 to 0.97 at 400 and 500 nm, respectively, for hazy days, and from 0.91 to 0.99 and 0.94 to 0.99 at 400 and 500 nm, respectively, but from 0.989 to 0.999 at 400 nm and from 0.990 to 0.999 at 500 nm under clean weather conditions, showing steady variation. The Ångström exponent varies from 0.04 to 0.88, 0.78 to
0.82, and 0.99 to 1.07 on clean, dusty and haze days, respectively, suggesting that on polluted days it varies smoothly while on clean days it fluctuates considerably. The volume size distribution under clean conditions presents a single-modal pattern, but under dusty and haze weather conditions it presents a typical bimodal pattern.

Measurements from the SKYNET skyradiometer can be used to analyze the AOPs over Beijing reasonably. In certain aspects, such as monitoring dust events, this instrument is likely to have better performance.

**Code/Data availability.** The AERONET data are available at https://aeronet.gsfc.nasa.gov/new_web/aerosols.html.
The SKYNET data used in the study can be requested by contacting the first author of the paper (xianyiyang7@163.com).

**Author contribution.** All authors contributed to shaping up the ideas and reviewing the paper. Xianyi Yang, Huizheng Che and Quanliang Chen designed the present study and prepared the manuscript; Hitoshi Irie, Xianyi Yang and Ying Cai
contributed to the SKYNET data retrieval; Linchang An, Hujia Zhao and Yu Zheng performed observation; Ke Gui, Xianyi



Yang, Lei Li and Yuanxin Liang contributed to analysis of the dataset; Yaqiang Wang, Hong Wang and Xiaoye Zhang provided constructive comments on this study.

**Competing interests.** The authors declare that they have no conflict of interest.

**Special issue statement.** This article is part of the special issue "SKYNET – the international network for aerosol, clouds, and solar radiation studies and their applications". It does not belong to a conference.


**Acknowledgements.** This research has been supported by the National Science Fund for Distinguished Young Scholars (grant no. 41825011), the National Key R & D Program Pilot Projects of China (grant nos. 2016YFA0601901 and 2016YFC0203304), the National Natural Science Foundation of China (grant no. 41590874), the CAMS Basis Research Project (grant no. 2017Z011), the European Union Seventh Framework Programme(FP7/2007-2013) (grant no. 262254), and

the AERONET-Europe ACTRIS-2 program, European Union's Horizon 2020 research and innovation programme (grant no. 654109).

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





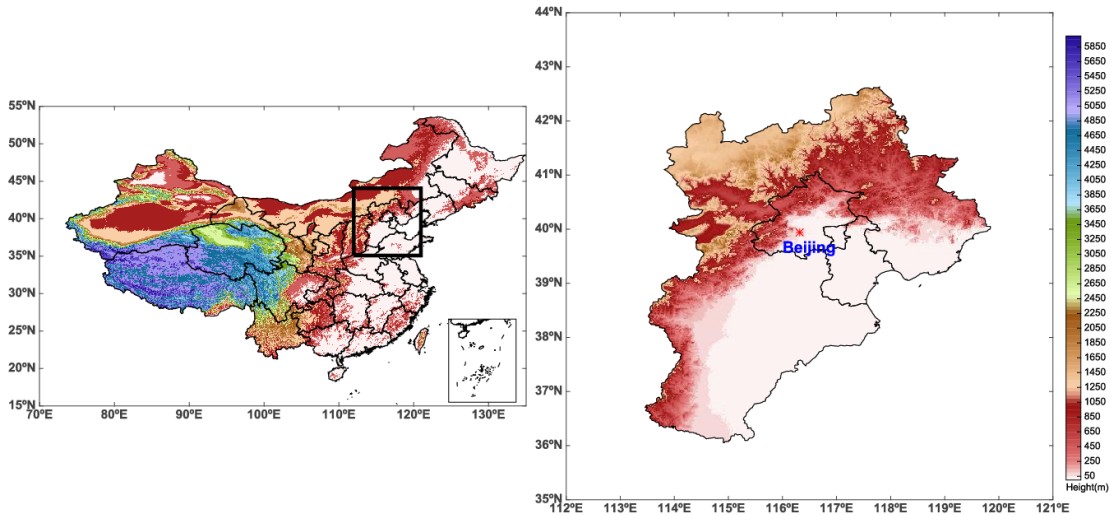

**Figure 1: Terrain elevation and location of the study site.**

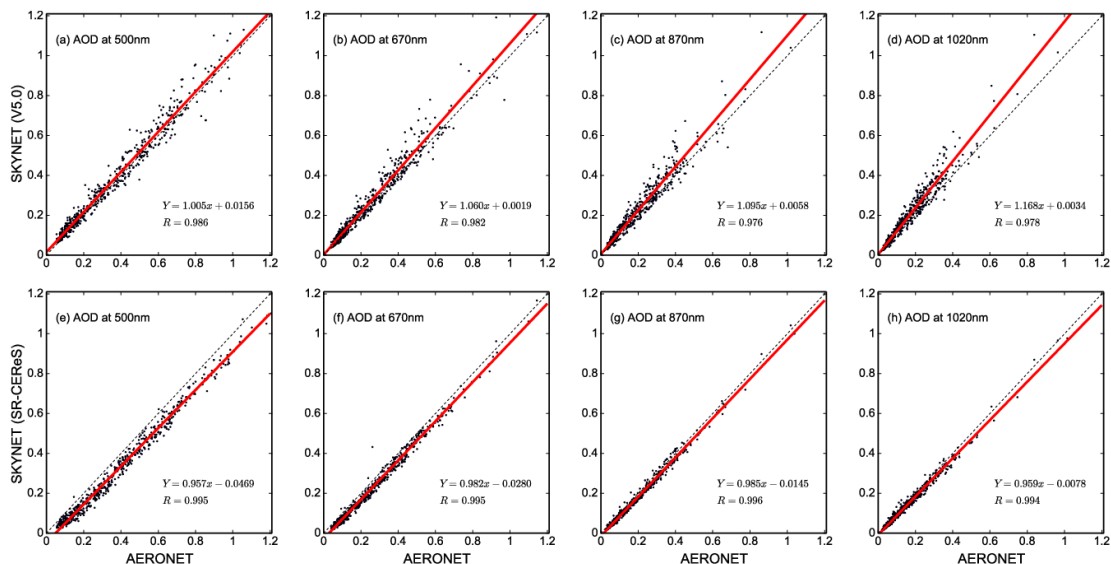


**Figure 2: Comparison of SKYNET SR-CEReS–retrieved and SKYNET V5.0–retrieved AOD with that from AERONET (within 1 minute) at 500, 670, 870 and 1020 nm over Beijing. The red solid line is the fitted linear regression curve.**





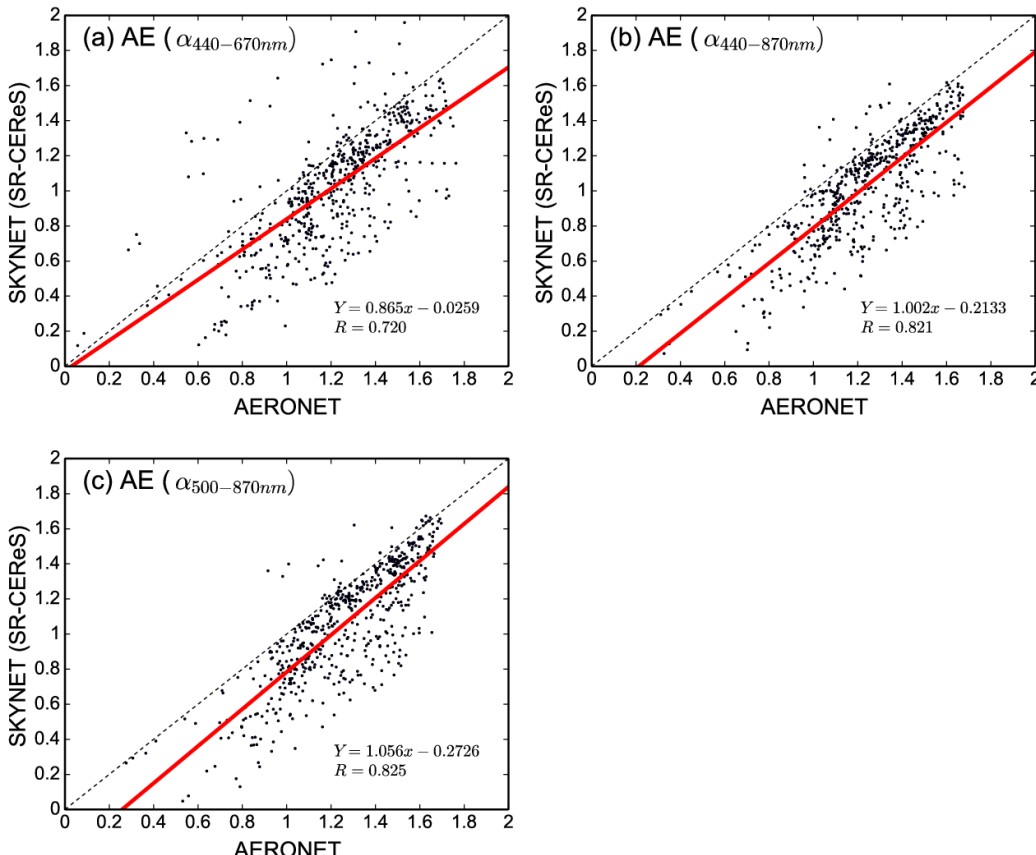

**Figure 3: Comparison of SKYNET with AERONET Ångström exponent (within 1 minute) at 440–670 nm, 440–870 nm and 500–870 nm over Beijing. The red solid line is the fitted linear regression curve.**





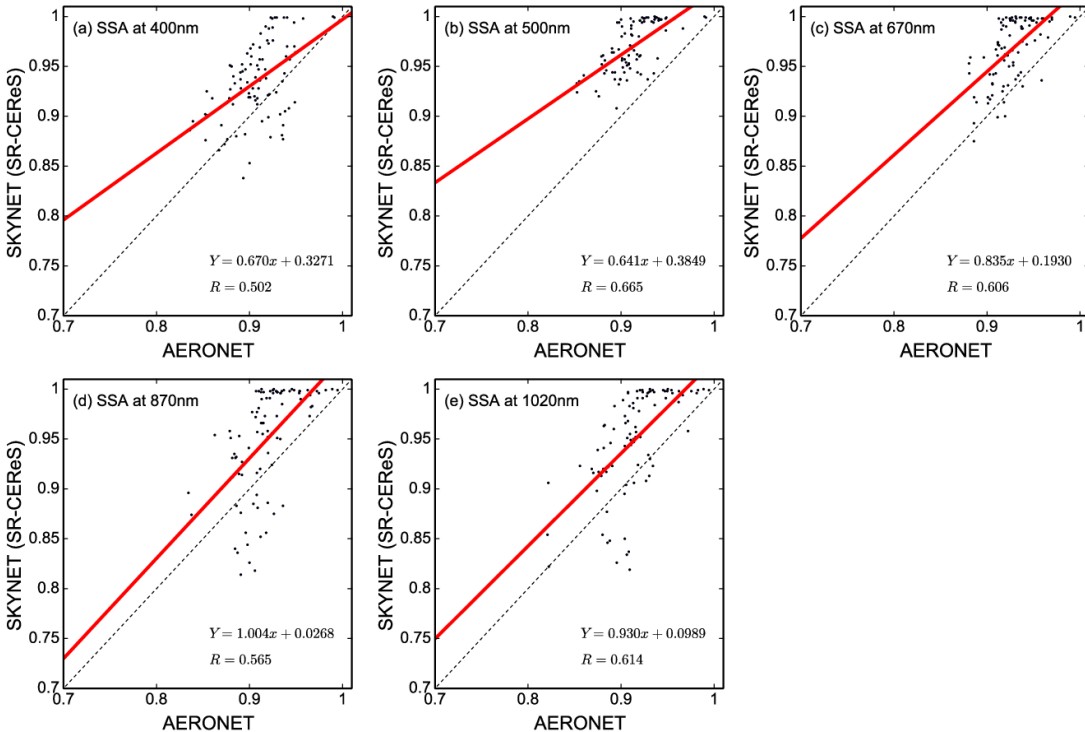

**Figure 4: Comparison of SKYNET and AERONET SSA (within 3 minutes) at 400, 500, 670, 870 and 1020 nm over Beijing. Only data with AOD > 0.4 are shown. The red solid line is the fitted linear regression curve.**





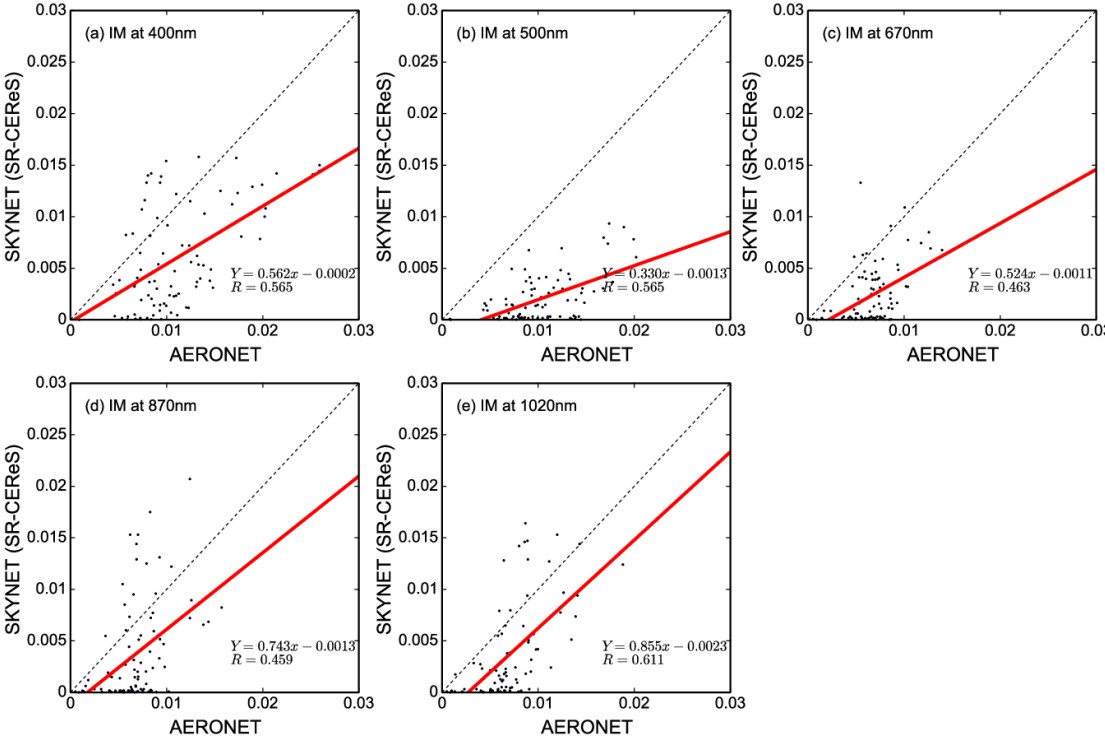

**Figure 5: Comparison of the SKYNET and AERONET imaginary part of the complex refractive index (within 3 minutes) at 400, 500, 670, 870 and 1020 nm over Beijing. Only data with AOD > 0.4 are shown. The red solid line is the fitted linear regression curve.**





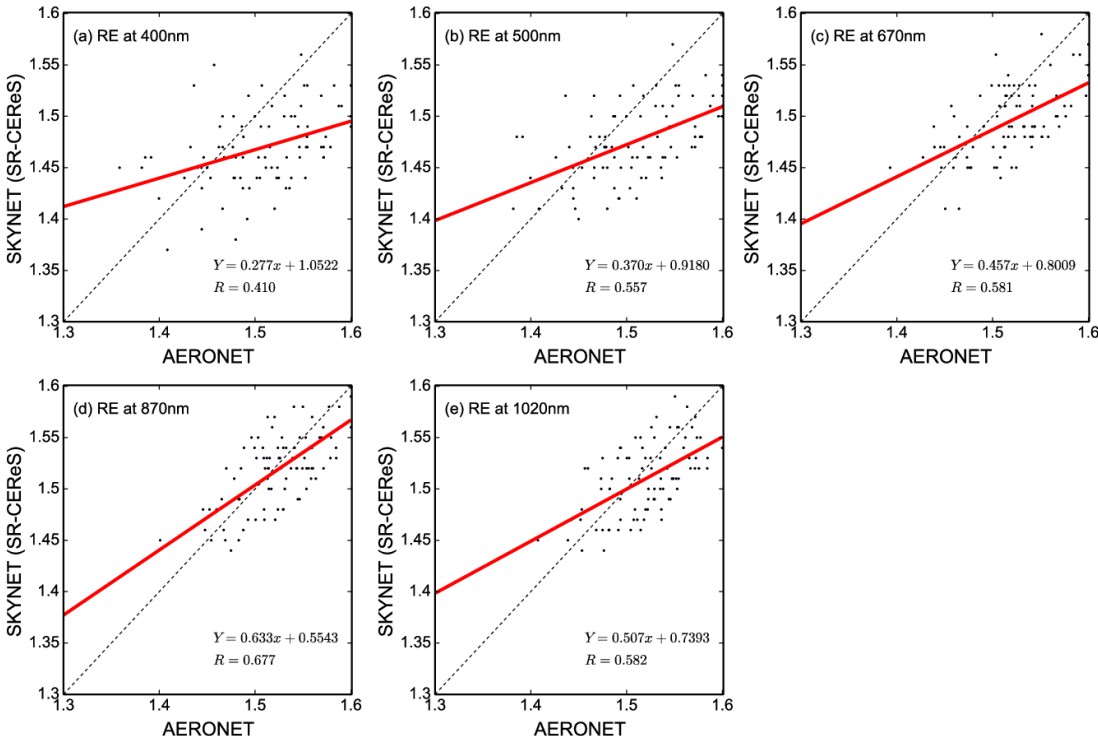

**Figure 6: Comparison of the SKYNET with AERONET real part of the complex refractive index (within 3 minutes) at 400, 500, 670, 870 and 1020 nm over Beijing. Only data with AOD > 0.4 are shown. The red solid line is the fitted linear regression curve.**


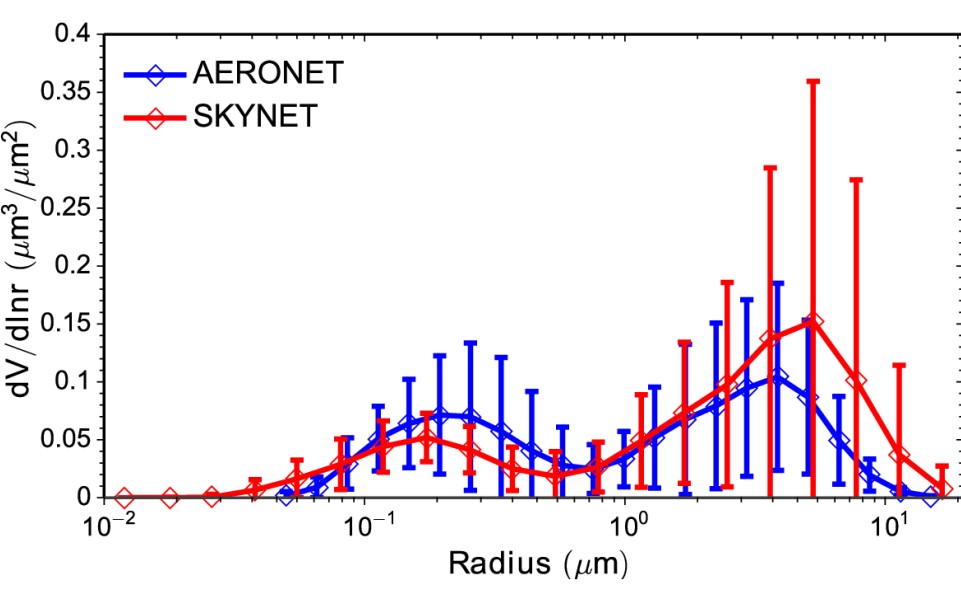





**Figure 7: Comparison of SKYNET- and AERONET-retrieved volume size distributions (within 3 minutes) over Beijing.**

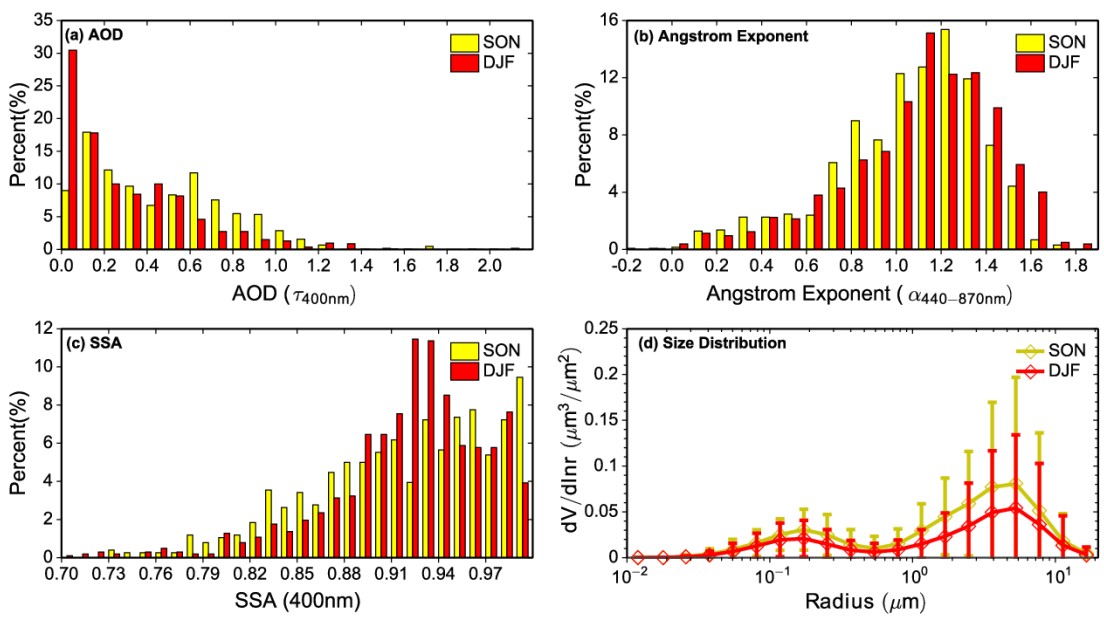

**Figure 8: Frequency distribution of (a) AOD at 400 nm, (b) Ångström exponent at 440–870 nm, (c) SSA at 400 nm, and (d) volume size distribution in autumn (September–October–November, SON) and winter (December–January–February, DJF) over Beijing for the period from September 2016 to January 2019.**

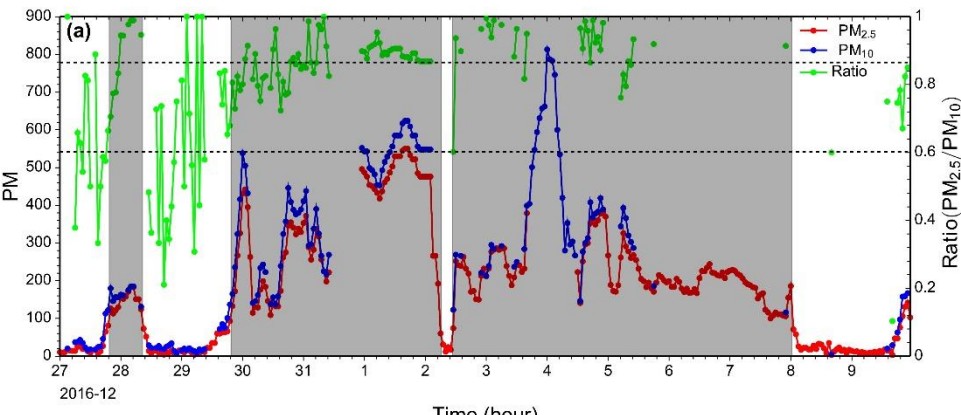





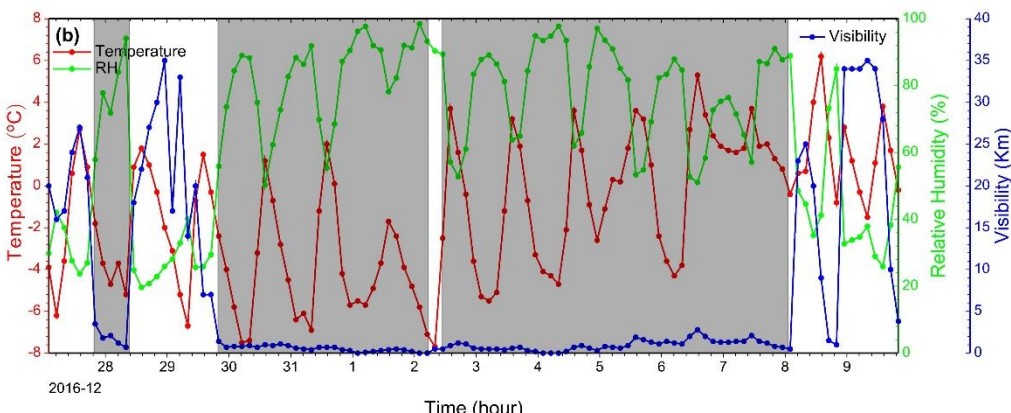


**Figure 9: Temporal variations of (a) PM2.5, PM10, and their ratio (PM2.5/PM10), and (b) meteorological data including temperature (°C), RH (%), and visibility (km), from 27 December 2016 to 9 January 2017. The shaded parts represent the pollution periods.**

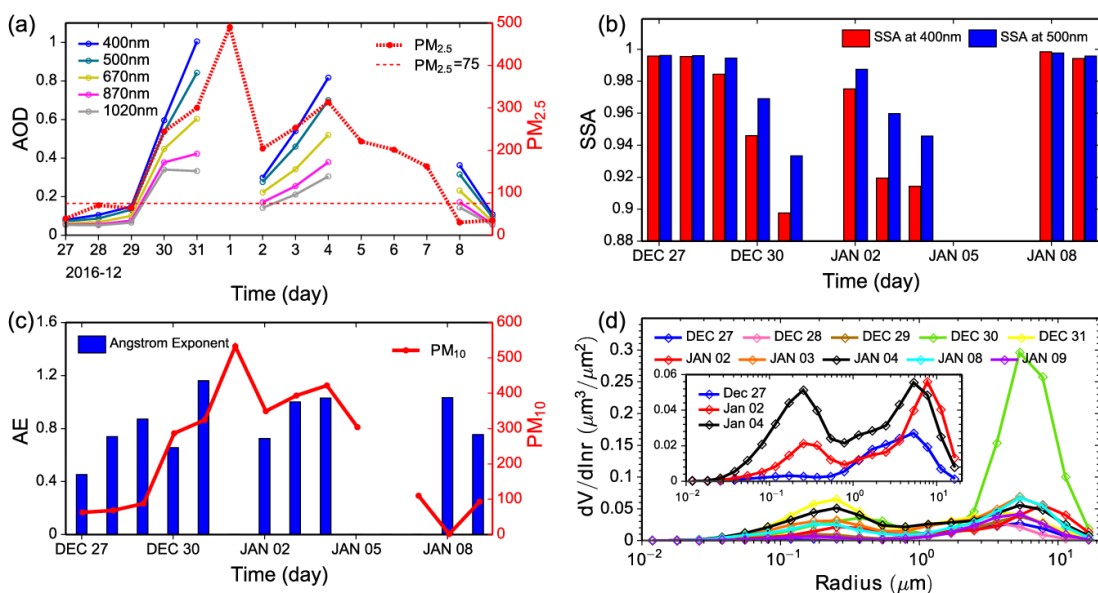


**Figure 10: Daily averaged variation in AOPs from SKYNET skyradiometer measurements in Beijing from 27 December 2016 to 9 January 2017: (a) AOD; (b) SSA; (c) Ångström exponent; and (d) volume size distribution.**





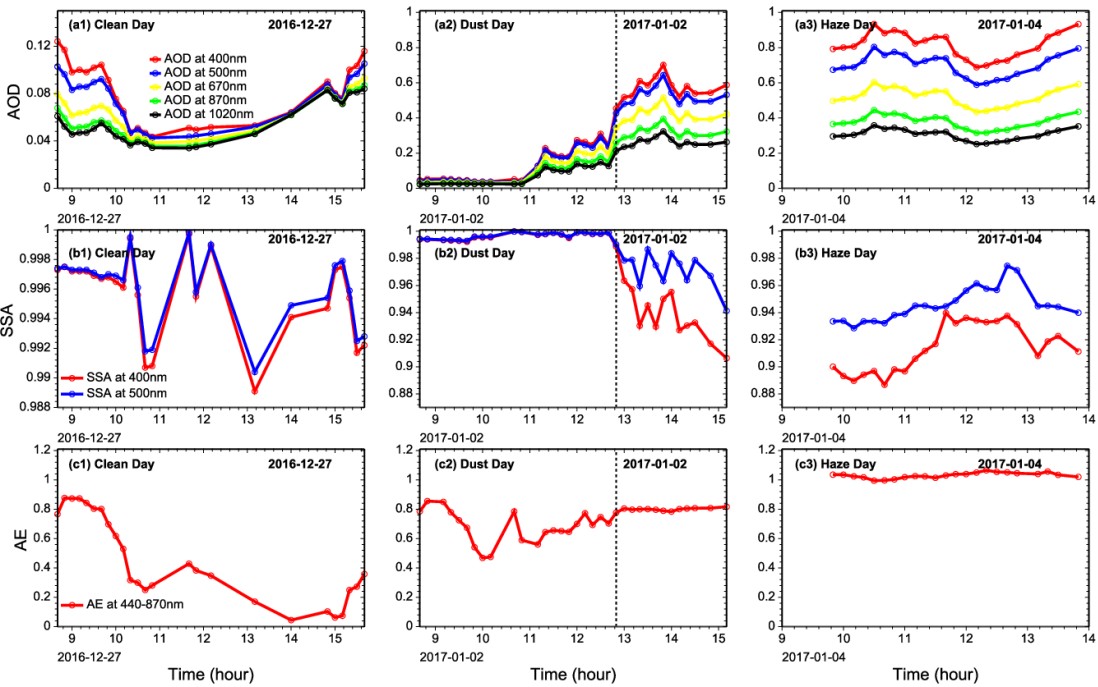

**Figure 11: Temporal variation of (a) AOD, (b) SSA, and (c) Ångström exponent from the SKYNET skyradiometer under (a1, b1, c1) clean, (a2, b2, c2) dusty, and (a3, b3, c3) hazy weather conditions in Beijing on 27 December 2016, 2 January, and 4 January 2017, respectively.**

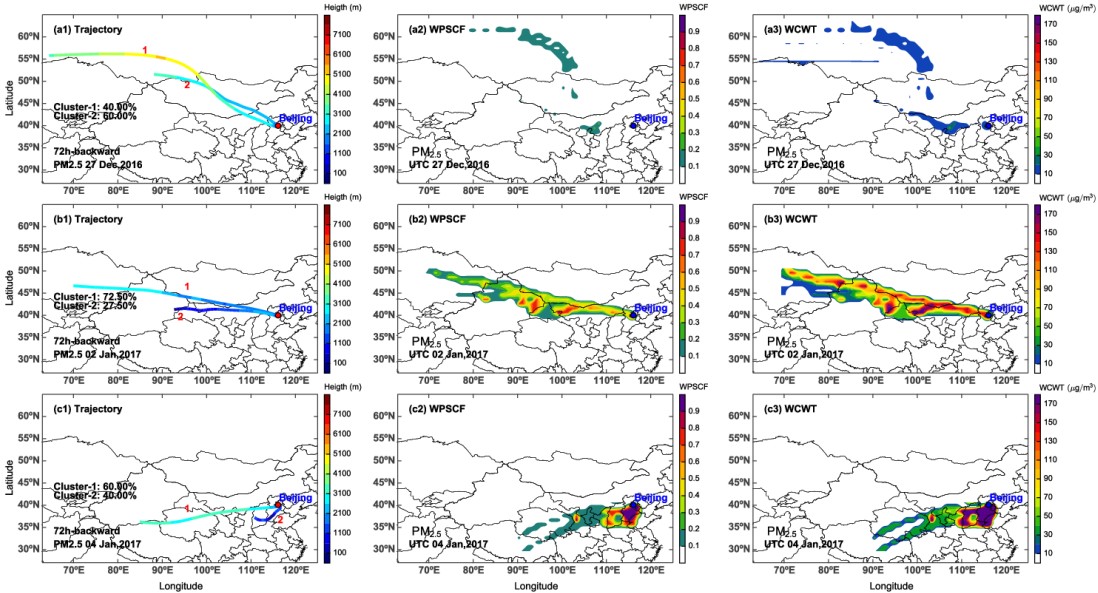

**Figure 12: Mean 72-hour backward trajectories of each trajectory cluster and spatial distribution of WPSCF and WCWT values for PM2.5 in Beijing during (a) 27 December 2016, (b) 2 January 2017, and (c) 4 January 2017.**