# Peer review of "Retrieval of aerosol optical properties over Beijing: a comparison between SKYNET and AERONET"

_Atmospheric Measurement Techniques, 2019_

## Referee Comment (RC1) · Anonymous Referee #1 · 9 Oct 2019

Title: Retrieval of aerosol optical properties over Beijing: a comparison between SKYNET and AERONET

Authors: Xianyi Yang, Huizheng Che, Hitoshi Irie, Quanliang Chen, Ke Gui, Ying Cai, Yu Zheng, Linchang An, Hujia Zhao, Lei Li, Yuanxin Liang, Yaqiang Wang, Hong Wang, Xiaoye Zhang

Review: Atmospheric Measurement Techniques

General Comments:

This paper overall contains some useful information on the SKYNET measurements

and retrievals. However, I feel the material does not meet the standards of an AMT publication since these is relatively little new information that is not already in the scientific literature and therefore does not present significant or substantial scientific progress in remote sensing of aerosols. I urge the authors to submit this paper to a different journal since there are some analyses that may provide some insight into aerosol properties in Beijing. Additionally, the paper is poorly written with numerous issues such as lack of clarity on dates utilized for analysis, poor statistical representativeness (only one pollution event and one dust event), incomplete or even erroneous descriptions of AERONET measurements and/or algorithms, and in some cases poor choices of comparison metrics. I provide numerous examples of these issues and other issues below in Specific Comments so that these may be improved in a revised submission.

Specific Comments:

Line 14, Abstract: Correlation coefficient is not the best way to compare AOD from these two networks. Better comparison metrics for AOD would be RMS differences and bias.

Line 16, Abstract: The SVA cannot be changed as it is a characteristic of the hardware of the instrument. I assume you might mean calibration adjustment therefore this should be reworded.

Line 17-19, Abstract: For size distributions it is more important to compare the modal or peak sizes of both the fine and coarse modes for the 2 retrievals rather than comparing the volume.

Line 17-19, Abstract: Be clear here on how many cases were utilized to compute these comparisons. It seems to me that the sample size of one pollution event and one dust event compared to one clear event is not nearly large enough to be statistically robust.

Line 45, Introduction: Please mention here that AERONET is global, as compared to the regional SKYNET.

Line 60, Introduction: This line about the Pandithurai et al. (2009) reference should be removed, it does not fit or make sense here.

Line 66-67, Introduction: Very poorly written, were all AOPs in the best agreement at 675 nm, including the AOD?

Line 68-69, Introduction: Why do you not provide a summary of the Estelles et al. (2012) results when you did for the other comparison papers that you referenced?

Line 78-79, Introduction: This is the wrong reference (Mok et al., 2016) for this analysis package. Note that in the first sentence of section 2.2 you give a web address for this package. This is an example of errors and lack of attention to detail in this manuscript.

Line 84, Section 2.1: Please be specific on the AERONET data used, did you analyze the Beijing-CAMS site data in Sep 2016 and the Beijing site data in March 2007? Also be clear here: are there only 2 months of data used in the comparisons between SKYNET and AERONET retrievals?

Lines 89-90, Section 2.1: This is a very poor description of the Cimel spectral measurements of most of the AERONET network. The AOD measurements at most sites also include 340, 380, 500 and 1640 nm. Also, the three channel 870 nm-channel polarization measurements were made at only a few sites in AERONET and retrievals did not use these channels. Additionally, you need to include uncertainty values of the AERONET data, such at the 0.01-0.02 for AOD (Eck et al. 1999) and the 0.03 for 440 nm SSA when AOD>0.5 (Dubovik et al., 2001).

Lines 99-100, Section 2.2: Again, an incomplete description here. The AERONET algorithm uses a mixture of both spherical and spheroidal particles, with the percentage spherical determined by the best fit to the measured sky radiances.

Line 104, Section 2.2: You should note that this results in an uncertainty in AOD ranging from 0.01 to 0.025 for overhead sun (optical airmass=1).

Lines 106-108, Section 2.2: It seems that you are retrieving SKYNET AOD from two

different methods. More detail on the differences in these methods needs to be provided.

Lines 125-129, Section 2.2: However, this does not take into account the fact that AOD versus wavelength is not linear in logarithmic coordinates (see Eck et al. 1999). This is particularly true in Beijing since the fine mode particle size is often relatively large when AOD is high, and needs to be discussed.

Line 138, Section 2.3: Explain why you chose 72 hours for back trajectories given that aerosol lifetime is typically 1 week (7 days).

Line 156, Section 3.1: AOD is not retrieved by AERONET, no inversion algorithm is used to determine AOD. It is measured from direct sun observations. Also, you should cite the reference of Giles et al. (2019) when discussing the V3 AOD data.

Line 162, Section 3.1: Please give the biases in units of AOD also (i.e. ∼0.02 etc.).

Lines 162-164, Section 3.1: The way this sentence is written it is impossible to tell which of the 2 different SKYNET AOD values you are discussing here.

Line 168, Section 3.1: Please give some specifics here about what is done to select data 'carefully' here.

Line 170, Section 3.1: Please state in the text what is correlated in Figure 3, AERONET versus SKYNET from the SR-CEReS retrievals?

Line 174, Section 3.1: It would be useful to know if the correlation increases for the data subset where AOD(440)>0.4, since the Angstrom Exponent is highly uncertain at low AOD.

Lines 179-180, Section 3.1: You should interpolate the SKYNET 400 nm and 500 nm SSA to 440 nm by linear interpolation in linear coordinates, before making comparisons to AERONET.

Line 182, Section 3.1: Please also provide the bias in units of SSA (i.e. ∼0.03 differ-

ences).

Line 183, Section 3.1: Please note here that the SKYNET SSA retrievals hit the saturation value of 1.0 in a significant number of cases while AERONET SSA does not.

Line 188, Section 3.1: You need to define SA here and explain briefly how it relates to influencing the magnitude of sky radiances.

Lines 198-199, Section 3.1: Please make it clearer that these constant values of surface reflectance for all sites is a very crude assumption. Also state in the text that AERONET uses geographically and seasonally varying surface reflectance values that are much more robust.

Lines 215-217, Section 3.1: This is obvious, and you should mention that the imaginary refractive index is the retrieved parameter and that SSA is derived from this information along with the size distribution and the real refractive index.

Lines 225-227, Section 3.1: It would be useful if you provided some comparisons of volume size distributions of fine mode dominated (AE(440-870)>1.2) and also coarse mode dominated cases (AE<0.6).

Line 238, Section 3.1: These are typically called volume concentrations not volume spectra.

Line 247-251, Section 3.1: I think your earlier analysis of the effects of surface albedo and solid view angle are the more likely reasons for differences in SSA and IM between the AERONET and SKYNET retrievals. I suggest removing this sentence.

Line 251-252, Section 3.1: Some of this fine mode peak volume concentration variance is due to changes in fine mode radius from low AOD to high AOD conditions and also from dry to humid conditions. Showing one average size distribution is a very poor way to compare these two products. A scatterplot of SKYNET versus AERONET would be much more informative.

Line 254-255, Section 3.1: This is by definition since these are the limits of the size distribution for AERONET and there is a strong constraint in the AERONET retrieval that results in near zero values at the limits.

Line 260-265, Section 3.2: Please make it clear in the figure captions that these frequency distributions are for the SKYNET retrievals only.

Line 266-271, Section 3.2: Are these frequency distributions from multi-year data? Or just one year for each season? The AERONET data base would provide much more robust statistics for fall versus winter than just these 2 seasons of data from SKYNET. I suggest remaking the frequency distribution plots from multi-year AERONET data in order to have a more robust seasonal comparison, including for all four seasons.

Line 280-281, Section 3.2: Similar comment as above: Since the study period of the SKYNET data is so limited it seems that you cannot make any general statements about fall versus winter aerosol properties unless you redo this climatological analysis with AERONET data and expand it to all 4 seasons.

Line 284-286, Section 3.2: It is well known that the major dust season in this region is spring and you ignore this since the SKYNET data set is so limited.

Line 288-289, Section 3.2: Hygroscopic particle growth leads to larger fine mode particles, not a shift from fine to coarse mode, as this sentence suggests.

Line 292, Section 3.3: Figure 9 x-axis labels are wrong, it is not hours but days, from Dec 27 through Jan 9, 2017.

Line 296, Section 3.3: Define the ratio here in the text as PM2.5/PM10.

Line 302, Section 3.3: You should also discuss the large amount of fog present from Jan 1 - Jan 5, 2017 in the region around Beijing (MODIS images show this), and the high cloud cover amount on Jan 4, 2017. Li et al. (2014) in Atmospheric Environment analyze the high AOD and PM associated with fog in Beijing in January of 2013. Fog may be associated with large secondary production of aerosols in the region. Also, Eck

et al., 2018 show the association of high cloud amount in Beijing with high AOD levels, likely due to humidification growth of fine particles.

Line 319-320, Section 3.3: You show a PM2.5 value in Figure 10 for Jan 4 but in Figure 9 you only show the PM10 value for that day. This inconsistency needs to be explained or else corrected.

Line 371, Section 3.3: The AOD levels are too low on the clean day to have reasonably accurate SSA retrievals. You should not even discuss such SSA values that would have very large uncertainties (~0.10).

Line 391-392, Section 3.3: Please explain the trajectory # 1 and #2 as shown in the Figure 12.

Line 405, Section 3.3: All 3 days show 2 clusters of trajectories although the differences in these are not explained. Are they from different altitudes?

Line 420, Section 4: Add 'of absorption ' here, after 'weakening'.

Line 425-426, Section 4: This may be true for past data sets, but the newest AERONET instruments now take hybrid scans hourly that provide more frequent retrievals throughout the entire day.

Line 427-428, Section 4: This sentence does not make any sense in the context of this paragraph and therefore should be removed.

Line 435-438, Section 5: In addition to the correlation coefficient is it import to also provide the rms differences and the average bias for the AOD and AE plus refractive indices (both real and imaginary) comparisons.

Line 454-455, Section 5: It was never explained why only autumn and winter were analyzed, and the other 2 seasons ignored in this paper.

Line 462, Section 5: Again, poor writing, since Dec 27 is a clear day, Jan 2 is a dust event and Jan 4 is a haze event, yet you identify the whole two-week interval as a haze

event.

Line 468-475, Section 5: The way this is written it implies some statistics of many days of data in each category of clear, dusty and hazy. In actuality this is only one day of each type and therefore these numbers of very limited value.

Line 478-479, Section 5: These sentences should be removed since they are vague and it was not clearly shown in the paper why the SKYNET retrievals will have better performance for dust events than for haze events.
* * *

---

## Referee Comment (RC2) · Anonymous Referee #2 · 29 Oct 2019

This study consists on two different pieces of work. First, a comparison of the aerosol optical properties retrieved by SKYNET and AERONET methodologies, based on data obtained at Beijing. Second, a case study in Beijing using the SKYNET data only, for a limited period.

General comments: the article has some interest given the importance of the comparisons between AERONET and SKYNET methodologies, from the point of view of homogeneity between networks. The authors list a number of previous papers devoted to comparisons between both networks. In this case, the comparison is made between AERONET version 3, and SKYRAD version 5 implemented in the SR-CEReS package, so the results are to some extent, new. However, I think there are some flaws that should be addressed before this paper was accepted for publication in this journal: First, the comparison should be improved by performing a more in deoth analysis of the retrieval differences; second, the example analysis of the episode should be discussed in the light of the different methods; or alternatively, removed or shortened, as to give emphasis to the comparison itself.

Abstract: the analysis of the winter episode has too much weight given the article title. I would expect to focus the paper more on the comparison itself.

Line 94-96: Given the different versions co-existent, it would be good to make clear the current choices available. In this sense, the original skyrad version was not 4.2 but previous.

Line 161: if the differences are given in %, then the definition of the MBD cannot be (mean_aeronet - mean_skynet). Please define properly. Do you mean you first compute the mean value for all the aeronet and skynet datasets separately and then compare with the MBD? Or do you compute the differences for every pair of coincident data, and then perform the mean? The discussion of the correlation coefficient is not enough so I would recommend to include other statistical parameters such as RMS.

Lines 170-174: the discussion about the Angstrom exponent should be analysed in more depth, for example the exponent is highly uncertain for low AOD. This should be taken into account in the analysis and/or discussion.

Line 189: the sensitivity tests performed are of interest for SKYNET users. I think the authors should give more emphasis on these results than in the analysis of the episode. How could the comparison improve by finding an effective SVA? Would be the results consistent with Kathri et al 2016?

Figures 4 and 5: values of SKYRAD lying on the 1 and 0 axis are probably due to by-default values in case of good retrievals and should perhaps be removed.

[Figure]

Line 221: I don't think the linear correlations are clear. The deviation of points relative to the fitted line is high.

Line 225: the comparison of size distributions could also be studied with more detail, for example depending on the radius bands, or depending on the type of aerosol present. Comparison to nominal uncertainties would be informative.

Section 3.2: I understand to include the analysis of an episode to demosntrate the uselfulness of the SKYNET retrievals, however, I think more effort should be put in the comparison than in the example analysis. Possibly a comparison with the AERONET data during the episode should be included too.

Line 301: negation correlation should be negative correlation

Section 4. The discussion section looks redudant. It should be included in the results or conslucions.

---

## Author Comment (AC1) · 13 Jan 2020

We appreciate the reviewer' valuable comments and constructive suggestions, which help improve the quality of the manuscript. We have carefully revised the manuscript according to these comments. The reviewer's comments are in black, our responses are in blue and the corresponding changes in manuscript are in red.

**Anonymous Referee #1**

General Comments: This paper overall contains some useful information on the SKYNET measurements and retrievals. However, I feel the material does not meet the standards of an AMT publication since these is relatively little new information that is not already in the scientific literature and therefore does not present significant or substantial scientific progress in remote sensing of aerosols. I urge the authors to submit this paper to a different journal since there are some analyses that may provide some insight into aerosol properties in Beijing. Additionally, the paper is poorly written with numerous issues such as lack of clarity on dates utilized for analysis, poor statistical representativeness (only one pollution event and one dust event), incomplete or even erroneous descriptions of AERONET measurements and/or algorithms, and in some cases poor choices of comparison metrics. I provide numerous examples of these issues and other issues below in Specific Comments so that these may be improved in a revised submission.

**Response:** Thank you for your valuable comments and constructive suggestions. We have modified the expressions in a proper way. The comparison metrics include RMSE, MBD and correlation coefficient in the revised manuscript. The frequency distribution of AERONET-retrieved AOPs has given in the revised manuscript comparing with SKYNET-retrieved AOPs for the four seasons. We have added the AERONET data in the pollution event analysis to compare with SKYNET data, and the poor statistical results (only one pollution event and one dust event) has removed. The comparison of AERONET and SKYNET data on the three days are shown instead. Some detail responses are in the following:

**Specific Comments:**

Line 14, Abstract: Correlation coefficient is not the best way to compare AOD from these two networks. Better comparison metrics for AOD would be RMS differences and bias.

**Response:** Following the reviewer's comment, the RMSE and MBD have added in the revised manuscript. (Line 14-18)

'The results obtained from simultaneous measurements compare well (RMSE of 0.010-0.020) and show high correlation coefficients (> 0.996) for aerosol optical depth (AOD) at each wavelength. The highest correlation coefficient for Ångström exponent (when  $AOD_{440nm}$ >0.4) is 0.992, at 440–870 nm, with the smallest RMSE of 0.042. The RMSE of single scattering albedo (SSA) between SKYNET and AERONET is as low as 0.018 at 440 nm, with high correlation coefficient (0.851), and adjusting the sky-radiance calibration constant and surface albedo input values can easily affect the value of SKYNET SSA. The real and imaginary parts of the refractive index show deviations of 0.031-0.055 and 0.003-0.005 respectively for all the wavelengths.'

Line 16, Abstract: The SVA cannot be changed as it is a characteristic of the hardware of the instrument. I assume you might mean calibration adjustment therefore this should be

**reworded.**

**Response:** We have replaced 'SVA' with 'sky-radiance calibration constant'. (Line 17) 'while the sky-radiance calibration constant and surface albedo input values can affect the value of SKYNET SSA.'

Line 17-19, Abstract: For size distributions it is more important to compare the modal or peak sizes of both the fine and coarse modes for the 2 retrievals rather than comparing the volume. **Response:** The comparison of the fine and coarse modes has added, and the appropriate description about the comparison results has shown in the revised manuscript. (Line 19-22) 'The fine mode and coarse mode dominated volume size distribution patterns derived from the two networks' instruments are both bimodal but the coarse-mode volume concentration in coarse mode dominated condition is much larger that in fine mode dominated, meanwhile the coarse-mode volume of SKYNET is larger than that of AERONET on average.'

Line 17-19, Abstract: Be clear here on how many cases were utilized to compute these comparisons. It seems to me that the sample size of one pollution event and one dust event compared to one clear event is not nearly large enough to be statistically robust. **Response:** Following the reviewer's comment, the revised manuscript now has modified the sentences. We have shown a clear date in the description of the statistics. (Line 31-34) 'The AOD shows high consistency for the SKYNET skyradiometer and AERONET suphotometer on the clean day (27 December 2016), light-pollution condition (2 January 2017) and heavy-pollution condition (4 January 2017), and the RMSE are about 0.005, 0.006 and 0.018 respectively. The RMSE of SSA are 0.022, 0.046, 0.020 and for Ångström exponent are 0.229, 0.289, 0.060, respectively, the large biases of Ångström exponent are due to the low AOD values.'

Line 45, Introduction: Please mention here that AERONET is global, as compared to the regional SKYNET.

**Response:** It has mentioned in the sentence in the revised manuscript. (Line 47-49)

'Ground-based measurement networks are a very useful and accurate way to monitor the spatiotemporal distribution of aerosols (Holben et al., 2001) by using the sun-sky radiometric technique (Holben et al., 1998). SKYNET (Nakajima et al., 2007; Takamura and Nakajima, 2004), located mostly in Asia and Europe, is a regional observation network dedicated to aerosol–cloud–radiation interaction research (Nakajima et al., 1996; Nakajima et al., 2007; Che et al., 2014). AERONET (Aerosol Robotic Network; Holben et al., 2001) is a global well-known ground-based remote-sensing aerosol network, established by NASA (National Aeronautics and Space Administration) and PHOTONS (Photométrie pour le Traitement Opérationnel de Normalisation Satellitaire).'

Line 60, Introduction: This line about the Pandithurai et al. (2009) reference should be removed, it does not fit or make sense here.

**Response:** This line about the Pandithurai et al. (2009) reference has removed in the revised manuscript. (Line 63)

Line 66-67, Introduction: Very poorly written, were all AOPs in the best agreement at 675 nm, including the AOD?

**Response:** Sorry, it should be AODs. The sentence has modified in the revised manuscript. (Line 69)

'Evgenieva et al. (2008) compared the AODs between AERONET and SKYNET using two days of measurements and found the lowest deviation to be at 675 nm.'

Line 68-69, Introduction: Why do you not provide a summary of the Estelles et al. (2012) results when you did for the other comparison papers that you referenced?

**Response:** The more detail description has added in the sentence in the revised manuscript. (Line 73-74)

'Estellés et al. (2012) then compared the differences between AERONET and SKYRAD4.2 inversion products retrieved from one month of Cimel data showing RMS differences of 0.025-0.049 for SSA, 0.005-0.034 and 0.004-0.007 for the real and imaginary parts of the refractive index, respectively.'

Line 78-79, Introduction: This is the wrong reference (Mok et al., 2016) for this analysis package. Note that in the first sentence of section 2.2 you give a web address for this package. This is an example of errors and lack of attention to detail in this manuscript.

**Response:** The wrong reference has removed in the revised manuscript. (Line 84)

Line 84, Section 2.1: Please be specific on the AERONET data used, did you analyze the Beijing-CAMS site data in Sep 2016 and the Beijing site data in March 2007? Also be clear here: are there only 2 months of data used in the comparisons between SKYNET and AERONET retrievals?

**Response:** Sorry for misunderstanding of previous statements, it has modified in the revised manuscript. The AERONET and SKYNET data sets from September 2016 to April 2018 are used for Section 3.1; while two networks' data sets from September 2016 to January 2019 are used for Section 3.2. (Line 90-91)

'The two instruments` measurements from September 2016 to April 2018 are used to analyze the differences between SKYNET and AERONET.'

Lines 89-90, Section 2.1: This is a very poor description of the Cimel spectral measurements of most of the AERONET network. The AOD measurements at most sites also include 340, 380, 500 and 1640 nm. Also, the three channel 870 nm-channel polarization measurements were made at only a few sites in AERONET and retrievals did not use these channels. Additionally, you need to include uncertainty values of the AERONET data, such at the 0.01-0.02 for AOD (Eck et al. 1999) and the 0.03 for 440 nm SSA when AOD>0.5 (Dubovik et al., 2001).

**Response:** According to the advice, the description of Cimel sunphotometer has modified, the uncertainty values of AOD and SSA have added in the revised manuscript. (Line 95-99) 'The sunphotometer is another automatic instrument for tracking the sun and scanning the sky, but with a 1.2° full field-of-view at the following channels: 340, 380, 440, 500, 675, 870, 940, 1020, and 1640 nm (Holben et al., 1998). Accuracy on AERONET AOD is 0.01-0.02 (Eck

et al. 1999) and the uncertainty values of 440 nm SSA when AOD > 0.5 is 0.03 (Dubovik et al., 2001). In this study, AERONET data from four channels (440, 675, 870, and 1020 nm) and SKYNET data from five channels (400, 500, 670, 870, and 1020 nm) are used to retrieve AOPs over Beijing.'

Lines 99-100, Section 2.2: Again, an incomplete description here. The AERONET algorithm uses a mixture of both spherical and spheroidal particles, with the percentage spherical determined by the best fit to the measured sky radiances.

**Response:** A complete description following the advice has given in the revised manuscript. (Line 106-107)

'The AERONET algorithm uses a mixture of both spherical and spheroidal particles, with the percentage spherical determined by the best fit to the measured sky radiance.'

Line 104, Section 2.2: You should note that this results in an uncertainty in AOD ranging from 0.01 to 0.025 for overhead sun (optical airmass=1).

**Response:** A more detail description in the sentence has given in the revised manuscript. (Line 112)

'Campanelli et al. (2004) presented a new procedure for the *in situ* determination of the solar calibration constant, and the precision of the method—by testing a five-month dataset obtained from a Prede skyradiometer in Rome, Italy—was estimated to fall within 1%–2.5% depending on the wavelength for overhead sun (optical airmass = 1).'

Lines 106-108, Section 2.2: It seems that you are retrieving SKYNET AOD from two different methods. More detail on the differences in these methods needs to be provided.

**Response:** Following the reviewer's comment, the revised manuscript now states that "The SR-CEReS developed by Chiba University selected the input data for the ILP method more carefully than before, from the measurements taken in more than 1 month before and after the target day (to keep sufficient number of data points) using a stricter criterion of error in input data." Additionally, because the skyrad.pack version 5.0 is included in the SR-CEReS as the main program, it should be only one method. Because of this, we do not say that SR-CEReS has been improved over version 5.0. So we remove the comparison between the two SKYNET retrieved AOD and AERONET retrieved AOD. The comparison between SKYNET SR-CEReS-retrieved AOD and AERONET-retrieved AOD is shown in Fig.2. The sentences have modified in the revised manuscript. (Line 113-115)

'The SR-CEReS software developed by Chiba University selects the input data for the ILP method more carefully than before, from the measurements taken in more than 1 month before and after the target day (to keep sufficient number of data points) using a stricter criterion of error in input data.'

Lines 125-129, Section 2.2: However, this does not take into account the fact that AOD versus wavelength is not linear in logarithmic coordinates (see Eck et al. 1999). This is particularly true in Beijing since the fine mode particle size is often relatively large when AOD is high, and needs to be discussed.

Response: Following the reviewer's comment, the reference and the sentence' When AOD is

high, the fine mode particle size is often relatively large, particularly in Beijing with high aerosol burdens' have added into the revised manuscript. When 440nm AOD>0.4, the simultaneous AEs (AE(440-870)>1.2)from SKYNET and AERONET have a little large RMS differences(i.e. ~0.05 etc.); However, the simultaneous AEs (AE(440-870)

Figure 2: Comparison of SKYNET SR-CEReS–retrieved AOD with that from AERONET (within 1 minute) at 500, 670, 870 and 1020 nm over Beijing. The red solid line is the fitted linear regression curve.

Line 168, Section 3.1: Please give some specifics here about what is done to select data 'carefully' here.

**Response:** We think this sentence makes no sense here, so it has removed. Specifics is that the SR-CEReS selected the input data from the measurements taken in more than 1 month before and after the target day (to keep sufficient number of data points) using a stricter criterion of error in input data.

Line 170, Section 3.1: Please state in the text what is correlated in Figure 3, AERONET versus SKYNET from the SR-CEReS retrievals?

**Response:** The sentence has been modified in the revised manuscript. (Line 182) 'A comparison of Ångström exponent retrieved from SKYNET SR-CEReS and AERONET is shown in Fig. 3.'

Line 174, Section 3.1: It would be useful to know if the correlation increases for the data subset where AOD(440)>0.4, since the Angstrom Exponent is highly uncertain at low AOD.

**Response:** According to the advice, the correlation increases a lot for the data subset when AOD(440)>0.4. The new results are shown in Fig. 3. (Line 182-190)

'A comparison of Ångström exponent retrieved from SKYNET SR-CEReS and AERONET is shown in Fig. 3. Only data with AOD440nm>0.4 are shown, since AE is highly uncertain at low values of AOD and the comparison result is bad (0.182-0.334 for all the wavelengths). Figure 3a-c show that the AE from AERONET is systematically lower than that from SKYNET SR-CEReS, the MBD of AE at 440-670 nm ( $\alpha_{440-670nm}$ ), 440-870 nm ( $\alpha_{440-870nm}$ ) and 500-870 nm ( $\alpha_{500-870nm}$ ) is 0.063, 0.016 and 0.009; the RMSE at 440-670 nm, 440-870nm and 500-870 nm is 0.080, 0.042 and 0.048; the correlation coefficient at 440-670 nm, 440-870nm and 500-870 nm is 0.986, 0.992 and 0.990. Both the highest correlation coefficient of AE and the lowest RMSE of AE are at 440-870 nm. The simultaneous AE within one minute ( $\alpha_{440-870nm} > 1.2$ ) from SKYNET SR-CEReS and AERONET has large RMS differences about 0.060; However, the simultaneous AE ( $\alpha_{440-870nm}

---

## Author Comment (AC2) · 13 Jan 2020

We appreciate the reviewer' valuable comments and constructive suggestions, which help improve the quality of the manuscript. We have carefully revised the manuscript according to these comments. The reviewer`s comments are in black, our responses are in blue and the corresponding changes in manuscript are in red.

**Anonymous Referee #2**

This study consists on two different pieces of work. First, a comparison of the aerosol optical properties retrieved by SKYNET and AERONET methodologies, based on data obtained at Beijing. Second, a case study in Beijing using the SKYNET data only, for a limited period.

General comments: the article has some interest given the importance of the comparisons between AERONET and SKYNET methodologies, from the point of view of homogeneity between networks. The authors list a number of previous papers devoted to comparisons between both networks. In this case, the comparison is made between AERONET version 3, and SKYRAD version 5 implemented in the SR-CEReS package, so the results are to some extent, new. However, I think there are some flaws that should be addressed before this paper was accepted for publication in this journal: First, the comparison should be improved by performing a more in deoth analysis of the retrieval differences; second, the example analysis of the episode should be discussed in the light of the different methods; or alternatively, removed or shortened, as to give emphasis to the comparison itself.

**Response:** Thank you for your valuable comments and constructive suggestions. In the revised manuscript, we do not compare the different SKYNET AOD at all, because the skyrad.pack version 5.0 is included in the SR-CEReS as the main program, it should be only one method. The difference is that SR-CEReS selected the input data from the measurements taken in more than 1 month before and after the target day (to keep sufficient number of data points) using a stricter criterion of error in input data. The frequency distribution of AERONET-retrieved AOPs has given in the revised manuscript comparing with SKYNET-retrieved AOPs for the four seasons. We have added the AERONET data in the pollution event analysis to compare with SKYNET data, and the comparison of AERONET and SKYNET data on the three days (clean, light-pollution, heavy-pollution) are shown instead. Some detail responses are in the following:

Abstract: the analysis of the winter episode has too much weight given the article title. I would expect to focus the paper more on the comparison itself.

**Response:** Following the reviewer's comment, the revised manuscript now has compared the frequency distribution between SKYNET and AERONET, and a comparison with the AERONET data during the episode has been included. (Line 13-34)

'**Abstract.** This study assesses the performance of SKYNET in comparison to AERONET (Aerosol Robotic Network) for retrieving aerosol optical properties (AOPs) in Beijing, China. The results obtained from simultaneous measurements compare well (RMSE of 0.010-0.020) and show high correlation coefficients (> 0.996) for aerosol optical depth (AOD) at each wavelength. The highest correlation coefficient for Ångström exponent (when $AOD_{440nm} > 0.4$) is 0.992, at 440–870 nm, with the smallest RMSE of 0.042. The RMSE of single scattering albedo (SSA) between SKYNET and AERONET is as low as 0.018 at 440 nm, with high

correlation coefficient (0.851), and adjusting the sky-radiance calibration constant and surface albedo input values can easily affect the value of SKYNET SSA. The real and imaginary parts of the refractive index show deviations of 0.031-0.055 and 0.003-0.005 respectively for all the wavelengths. The fine mode and coarse mode dominated volume size distribution patterns derived from the two networks' instruments are both bimodal but the coarse-mode volume concentration in coarse mode dominated condition is much larger that in fine mode dominated, meanwhile the coarse-mode volume of SKYNET is larger than that of AERONET on average.

According to the frequency distribution of SKYNET and AERONET retrieved AOPs, consistent conclusions are that the relatively high AOD values often occur in spring and summer, coarser aerosol particles often present in spring and finer particles usually exist in winter, and there are more absorbent aerosol particles in winter while more scattering aerosol particles in summer and autumn. SKYNET data, combined with AERONET data, meteorological data, CALIPSO (Cloud–Aerosol Lidar and Infrared Pathfinder Satellite Observations) data, backward trajectories, and WPSCF (weighted potential source contribution function) and WCWT (weighted concentrated weighted trajectory) analyses are used to analyze a serious pollution event in winter over Beijing. The results suggest that it was not only affected by local emissions but also by regional transport. The AOPs under three weather conditions (clean, light-pollution, heavy-pollution) in Beijing are discussed. The AOD shows high consistency for the SKYNET skyradiometer and AERONET sunphotometer on the clean day (27 December 2016), light-pollution condition (2 January 2017) and heavy-pollution condition (4 January 2017), and the RMSE are about 0.005, 0.006 and 0.018 respectively. The RMSE of SSA are 0.022, 0.046, 0.020 and for Ångström exponent are 0.229, 0.289, 0.060, respectively, the large biases of Ångström exponent are due to the low AOD values.'

Line 94-96: Given the different versions co-existent, it would be good to make clear the current choices available. In this sense, the original skyrad version was not 4.2 but previous.

**Response:** The Skyrad.pack algorithms corresponding to Nakajima et al. [1996] and Hashimoto et al. [2012] are denoted by Skyrad.pack (version 4.2) and Skyrad.pack (version 5.0) , respectively. In this study we use the SR-CEReS analysis package, but the main program of the package is version 5.0. One SKYNET retrieval (SKYNET V5.0) has removed in the revised manuscript, because the skyrad.pack version 5.0 is included in the SR-CEReS as the main program, it should be only one method. Because of this, we do not say that SR-CEReS has been improved over version 5.0. So we remove the comparison between the two SKYNET retrieved AOD and AERONET retrieved AOD. The comparison between SKYNET SR-CEReS-retrieved AOD and AERONET-retrieved AOD is shown in Fig.2. (Line 101-105)

'Figure 2 shows the results of SKYNET AOD compared with AERONET AOD. Figures 2a–d show that the AOD retrieved from the AERONET sunphotometer, at all wavelengths, is systematically higher than that retrieved from the SKYNET skyradiometer, and the MBD (mean bias deviation), defined as $\text{MBD} = \bar{\Delta} = \frac{1}{n}\sum_{i=1}^{n}\left(\delta_{skynet,i} - \delta_{aeronet,i}\right)$ at 500 nm, 670 nm, 870 nm and 1020 nm is -0.014, -0.015, -0.008 and -0.006 respectively. The RMSE (root mean square error), defined as $\text{RMSE} = \sqrt{\frac{1}{n}\sum_{i=1}^{n}\left(\delta_{skynet,i} - \delta_{aeronet,i}\right)^2}$ at 500 nm, 670 nm, 870 nm

and 1020 nm is 0.020, 0.020, 0.011 and 0.010, respectively. The correlation coefficient of AOD between the SKYNET SR-CEReS retrieval and AERONET at each channel is larger than 0.996. These statistical parameters confirm that the two networks' instruments are highly consistent in their measurement of AOD. Importantly, the AOD from SKYNET at 670 nm correlates to the AOD at 675 nm from AERONET, which may lead to the relatively large differences. Additionally, the comparison of AOD at shorter wavelengths has larger biases than that at longer wavelengths.'

[Figure]

**Figure 2: Comparison of SKYNET SR-CEReS–retrieved AOD with that from AERONET (within 1 minute) at 500, 670, 870 and 1020 nm over Beijing. The red solid line is the fitted linear regression curve.**

Line 161: if the differences are given in %, then the definition of the MBD cannot be (mean_aeronet - mean_skynet). Please define properly. Do you mean you first compute the mean value for all the aeronet and skynet datasets separately and then compare with the MBD? Or do you compute the differences for every pair of coincident data, and then perform the mean? The discussion of the correlation coefficient is not enough so I would recommend to include other statistical parameters such as RMS.

**Response:** The proper definition of MBD and RMS have shown in the revised manuscript. (Line 174-176)

'the MBD (mean bias deviation), defined as $\text{MBD} = \bar{\Delta} = \frac{1}{n}\sum_{i=1}^{n}(\delta_{skynet,i} - \delta_{aeronet,i})$ at 500 nm, 670 nm, 870 nm and 1020 nm is -0.014, -0.015, -0.008 and -0.006 respectively. The

RMSE (root mean square error), defined as $\text{RMSE} = \sqrt{\frac{1}{n}\sum_{i=1}^{n}(\delta_{skynet,i} - \delta_{aeronet,i})^2}$ at 500 nm, 670 nm, 870 nm and 1020 nm is 0.020, 0.020, 0.011 and 0.010, respectively.'

Lines 170-174: the discussion about the Angstrom exponent should be analysed in more depth, for example the exponent is highly uncertain for low AOD. This should be taken into account in the analysis and/or discussion.

**Response:** When AOD440nm>0.4, the comparison result between the AERONET and SKYNET is better than before. The more analysis about the AE are given in the revised manuscript. (Line 182-190)

'A comparison of Ångström exponent retrieved from SKYNET SR-CEReS and AERONET is shown in Fig. 3. Only data with $AOD_{440nm}$>0.4 are shown, since AE is highly uncertain at low values of AOD and the comparison result is bad (0.182-0.334 for all the wavelengths). Figure 3a-c show that the AE from AERONET is systematically lower than that from SKYNET SR-CEReS, the MBD of AE at 440-670 nm ($\alpha_{440-670nm}$), 440-870 nm ($\alpha_{440-870nm}$) and 500-870 nm ($\alpha_{500-870nm}$) is 0.063, 0.016 and 0.009; the RMSE at 440-670 nm, 440-870nm and 500-870 nm is 0.080, 0.042 and 0.048; the correlation coefficient at 440-670 nm, 440-870nm and 500-870 nm is 0.986, 0.992 and 0.990. Both the highest correlation coefficient of AE and the lowest RMSE of AE are at 440-870 nm. The simultaneous AE within one minute ($\alpha_{440-870nm}$>1.2) from SKYNET SR-CEReS and AERONET has large RMS differences about 0.060; However, the simultaneous AE ($\alpha_{440-870nm}$ <0.8) from SKYNET SR-CEReS and AERONET has small RMS differences about 0.013.'

[Figure]

**Figure 3: Comparison of SKYNET with AERONET Ångström exponent (within 1 minute) at 440–670 nm, 440–870 nm and 500–870 nm over Beijing. Only data with AOD > 0.4 are shown. The red solid line is the fitted linear regression curve.**

Line 189: the sensitivity tests performed are of interest for SKYNET users. I think the authors should give more emphasis on these results than in the analysis of the episode. How could the comparison improve by finding an effective SVA? Would be the results consistent with Kathri et al 2016?

**Response:** According to our results, we find that SVA and SA can easily affect the SSA, and the SVA is more important because the effects on the SSA is more sensitive. It is same as Kathri et al 2016. In order to get more accurate calibration constant for sky radiance, the SVA should be calculated in a short period. Average all values to get SVA is not very good, remove some large error points can get more accurate SVA.

Figures 4 and 5: values of SKYRAD lying on the 1 and 0 axis are probably due to by-default values in case of good retrievals and should perhaps be removed.

**Response:** Thanks for your advice. The values of SKYNET SR-CEReS lying on the 1 and 0 axis have been removed. The new results are shown in Fig. 4.

[Figure]

**Figure 4: Comparison of SKYNET and AERONET SSA (within 3 minutes) at 440, 670, 870 and 1020 nm over Beijing. Only data with AOD > 0.4 are shown. The red solid line is the fitted linear regression curve.**

Line 221: I don't think the linear correlations are clear. The deviation of points relative to the

fitted line is high.

**Response:** The inaccurate description has been removed in the revised manuscript. (Line 245)

Line 225: the comparison of size distributions could also be studied with more detail, for example depending on the radius bands, or depending on the type of aerosol present. Comparison to nominal uncertainties would be informative.

**Response:** Thanks for the advice, following the reviewer's comment, the revised manuscript now modifies the comparison of volume size distribution between AERONET and SKYNET, the fine-mode dominated and coarse-mode dominated volume size distribution have added. (Line 247-280)

'Comparisons of the volume size distribution of fine mode dominated  ($\alpha_{440-870nm}$>1.2) and coarse mode dominated  ($\alpha_{440-870nm}$<0.6) between AERONET and SKYNET are shown in Fig. 7, wherein only those data observed within 5 minutes of each other were considered as simultaneous. The volume of aerosol for an air column of unit cross section is used to express the columnar volume spectrum ($dV/dlnr$), and the radius is in logarithmic form (Nakajima et al., 1996). There are differences in the assumptions of size distribution between the SKYNET and AERONET retrieval algorithms. The volume at each rated radius is calculated by averaging the values at that radius for both the SKYNET skyradiometer and the AERONET sunphotometer. However, the number of rated radii for SKYNET and AERONET is 20 and 22, respectively, meaning 20 rated radii (0.012, 0.018, 0.026, 0.038, 0.055, 0.081, 0.118, 0.173, 0.253, 0.370, 0.541, 0.791, 1.156, 1.691, 2.473, 3.617, 5.289, 7.734, 11.310 and 16.540  μm) are used to retrieve the volume size distribution for SKYNET and 22 rated radii (0.050, 0.066, 0.086, 0.113, 0.148, 0.194, 0.255, 0.335, 0.439, 0.576, 0.756, 0.992, 1.301, 1.708, 2.241, 2.940, 3.857, 5.061, 6.641, 8.713, 11.432 and 15.000μm) are used to retrieve the volume size distribution for AERONET. As is shown in Fig. 7a, the size distribution patterns of fine mode dominated from SKYNET and AERONET are both bimodal, which is typical, but the peak volumes bear some differences. Specifically, the two peak volumes from the SKYNET skyradiometer are at the radii of 0.173  μm and 5.289  μm, with columnar volume concentrations of 0.060 and 0.093  $\mu m^3/\mu m^2$; whereas, those from the AERONET sunphotometer are at radii of 0.148  μm and 3.857μm, with columnar volume concentrations of 0.063 and 0.075  $\mu m^3/\mu m^2$. From Fig. 7b we can see that, the size distribution patterns of coarse mode dominated from SKYNET and AERONET both show a bimodal pattern. The two peak volumes from the SKYNET skyradiometer are at the radii of 0.081  μm and 3.617  μm , with columnar volume concentrations of 0.075 and 0.632  $\mu m^3/\mu m^2$ ; whereas, those from the AERONET sunphotometer are at radii of 0.086  μm  and 3.857μm, with columnar volume concentrations of 0.092 and 0.561  $\mu m^3/\mu m^2$. The significant difference between Fig. 7a and Fig. 7b is that the coarse-mode volume concentration in coarse mode dominated condition is much larger than that in fine mode dominated condition. One can see is that the coarse-mode volume concentration of SKYNET is larger than that of AERONET on average, whereas, in contrast, the fine-mode volume of SKYNET is smaller than that of AERONET on average. The SSA is a ratio that describes the scattering ability of aerosol particles and, generally, coarse-mode particles have a larger scattering ability, meaning the SSA will be larger when there are many coarse-mode particles. The difference in volume size distribution between SKYNET and AERONET might be one reason why the SSA retrieved from SKYNET is larger than that retrieved from

AERONET. It can be clearly seen that the deviations of the columnar volume concentrations around the peak volumes are larger than for other volumes, which is the same for both the SKYNET skyradiometer and the AERONET sunphotometer. However, the deviations for the volume of fine-mode particles retrieved from AERONET are larger than those of SKYNET in most cases, which is due to changes in fine mode radius from low AOD to high AOD conditions and also from dry to humid conditions; whereas, for the volume of coarse-mode particles, the deviations are larger for SKYNET than AERONET. From Fig. 7 we can see that the columnar volume spectrum retrieved from AERONET is nearly 0 $\mu m^3/\mu m^2$ at the radii less than 0.050 $\mu m$ and more than 15.000 $\mu m$. This is by definition since these are the limits of the size distribution for AERONET and there is a strong constraint in the AERONET retrieval that results in near zero values at the limits.

[Figure]

**Figure 7: Comparison of SKYNET- and AERONET-retrieved volume size distributions (within 5 minutes) of (a) fine mode dominated (AE(440-870)>1.2) and (b) coarse mode dominated cases (AE(440-870)<0.6) over Beijing.'**

Section 3.2: I understand to include the analysis of an episode to demosntrate the uselfulness of the SKYNET retrievals, however, I think more effort should be put in the comparison than in the example analysis. Possibly a comparison with the AERONET data during the episode should be included too.

**Response:** Thanks for the advice, following the reviewer's comment, a comparison with the AERONET data during the episode have added. (Section 3.3)

'Figure 11a shows the daily averaged 440 nm AOD variations from the SKYNET skyradiometer measurements and the AERONET sunphotometer measurements, and the daily averaged PM$_{2.5}$ during the study period. The missing values of AOD were caused by the accumulation of cloud. It can be clearly seen that the trend of AOD is similar to that of PM$_{2.5}$, and the daily averaged AOD values of SKYNET and AERONET are similar in Beijing. From 27 December 2016 to 29 December 2016, the SKYNET and AERONET AOD show a slow growth trend, but the values are lower than 0.2. On 30 December 2016, the AOD increases to 0.67 and 0.57 for SKYNET and AERONET, respectively. According to the volume size distribution on that day (Fig. 11d), the coarse mode dominates, with a peak volume concentration of 0.362 $\mu m^3/\mu m^2$ at the radius of 5.289 $\mu m$. Compared to the other days, the highest coarse-mode volume concentration indicates that there are dust aerosol particles over Beijing area. As can be seen from Fig. 11c, the SKYNET Ångström exponent is lower than 0.8 and the AERONET Ångström exponent decreased on 30 December 2016, which suggests that the proportion of coarse aerosol particles in the air is very large. Moreover, the instantaneous ratio is as low as 0.72, indicating a short-lived dust event happened over Beijing. The CALIPSO satellite can show

the vertical variation of aerosol subtypes (see Supplement), which can be used to distinguish these different subtypes, such as marine, dust, and polluted dust aerosol (Omar et al., 2009; Tao et al., 2014). Figure S5 shows that was a heavy aerosol layer under 3 km, including dust, smoke and polluted dust, near Beijing (39.93°N, 116.32°E), which helps explain the much higher volume concentration of coarse-mode particles on that day. According to the volume size distribution during 27 December 2016 to 31 December 2016, the peak fine-mode volume concentration of SKYNET increases gradually from 0.005 $\mu m^3/\mu m^2$ to 0.066 $\mu m^3/\mu m^2$ with the AERONET one increasing gradually from 0.006 $\mu m^3/\mu m^2$ to 0.089 $\mu m^3/\mu m^2$, which might also be related to the development of a haze event. The SSA reflects the effectiveness of aerosol scattering in the total extinction, which is one of the most important variables in assessing the influences of aerosols on the radiation budget (Khatri et al., 2016). As shown in Fig. 11b, the daily averaged SSA between SKYNET and AERONET show a same variation trend during these days. Owing to the accumulation of absorbing aerosol particles such as black carbon aerosols, which have strong absorption abilities, the SSA shows a downward trend from 30 December 2016 to 31 December 2016. Based on the variation trend of PM$_{2.5}$ values and the meteorological data, we can see that the air quality temporarily improves on 2 January 2017, because cold air blows through the Beijing area. However, a more serious pollution event happened from 2 January to 4 January in 2017. The daily averaged SKYNET AOD increases sharply from 0.33 to 0.85 and the AERONET ones varies from 0.32 to 0.96. The daily averaged PM$_{2.5}$ value increases from 204.12 $\mu m^3/\mu m^2$ to 313.92 $\mu m^3/\mu m^2$; and the ratio is as high as 0.82, indicating that the proportion of PM$_{2.5}$ is higher and the fine-mode aerosol particles played a key role in the formation of this haze event. Meanwhile, the volume size distribution of SKYNET shows a gradual increasing trend of fine-mode peak volume concentration from 0.021 $\mu m^3/\mu m^2$ to 0.053 $\mu m^3/\mu m^2$, the AERONET one varies from 0.030 $\mu m^3/\mu m^2$ to 0.057 $\mu m^3/\mu m^2$, and the ratio continuously increases to 0.94 during these three days, indicating that the proportion of fine pollutants became greater. The value of Ångström exponent is almost larger than 0.8 in the heavy haze event, which is similar to the findings of Eck et al. (2005), Xia et al. (2007) and Zheng et al. (2017). The SSA values for SKYNET and AERONET are both at low level on 31 December 2016 and 3 January, which indicates there are many absorbing aerosol particles. Both SKYNET and AERONET AOD are as low as 0.15on 9 January 2017, indicating the end of this pollution period.

Figure 12a shows the temporal variation of AOD from the SKYNET skyradiometer measurements and the AERONET sunphotometer at 440 nm on clean day (27 December 2016), light-pollution condition (2 January 2017) and heavy-pollution condition (4 January 2017), respectively. The daily averaged AOD from SKYNET and AERONET are less than 0.12 on 27 December 2016, which can be treated as the background AOD of Beijing. It should be emphasized that only when AOD at 440 nm is greater than 0.4 on 2 January 2017 do we consider it as light pollution (the results of the calculation of the pollution data is based on the standard of this day). It can be clearly seen that the AOD is very close between SKYNET and AERONET at the same time under the three weather conditions. The RMSE of AOD within 1 minute between SKYNET and AERONET are 0.005, 0.006 and 0.018 on the clean day (27 December 2016), light-pollution condition (2 January 2017) and heavy-pollution condition (4 January 2017), respectively. It indicates that the significant consistency of AOD for SKYNET skyradiometer measurements and the AERONET sunphotometer.

A comparison among the daily variations in SKYNET and AERONET SSA at 440 nm on clean day (27 December 2016), light-pollution condition (2 January 2017) and heavy-pollution condition (4 January 2017) is depicted in Fig. 12b. We can find that the SSA of SKYNET has much more data than AERONET so that it can reflect the variation of one day in more detail. Because the number of daily measurements of sky radiance by the SKYNET skyradiometer was more than that of the AERONET sunphotometer, and thus it is an advantage for SKYNET to use SSA values to analyze the daily variation. The newest AERONET instruments now take hybrid scans hourly that providing more frequent retrievals throughout the entire day, which can make up the defect. The RMSE of SSA within 10 minutes between SKYNET and AERONET are 0.022, 0.046 and 0.020 on the clean day (27 December 2016), light-pollution condition (2 January 2017) and heavy-pollution condition (4 January 2017), respectively. The biases of SSA are lower when AOD is high.

The temporal variations of SKYNET and AERONET Ångström exponent between 440 and 870 nm on clean day (27 December 2016), light-pollution condition (2 January 2017) and heavy-pollution condition (4 January 2017) are shown in Fig. 12c. The RMSE of AE within 1 minute between SKYNET and AERONET are 0.229, 0.289 and 0.060 on the clean day (27 December 2016), light-pollution condition (2 January 2017) and heavy-pollution condition (4 January 2017), respectively. The large differences of AE between SKYNET and AERONET are due to the low AOD values (AOD<0.2). The high consistency of AE occur when AOD is as high as 0.4.The volume size distributions of aerosol particles retrieved from the SKYNET skyradiometer and AERONET sunphotometer are shown in Fig. 11d and Fig. 11e, respectively. Furthermore, the volume size distributions on clean day (27 December 2016), light-pollution condition (2 January 2017) and heavy-pollution condition (4 January 2017) are selected to show individually. On clean day (27 December 2016), the volume size distribution is a typical bi-modal pattern, which is similar to the two networks, indicating the proportion of coarse-mode particles is much larger, which results in the Ångström exponent being very low. There are some differences of volume size distribution between SKYNET and AERONET on 2 January 2017, which is due to lacking AERONET data in clean condition (AOD<0.1) on that day. Both SKYNET and AERONET volume size distributions demonstrate a classic bimodal pattern on 4 January 2017, the volume concentrations of fine-mode particles on 4 January 2017is larger than that on 27 December 2016 and 2 January 2017 which clearly indicates fine particles have an important influence on heavy-pollution condition (4 January 2017).'

[Figure]

Figure 11: Comparison of daily averaged variation in AOPs from SKYNET and AERONET measurements in Beijing from 27 December 2016 to 9 January 2017: (a) volume size distribution of SKYNET; (b) volume size distribution of AERONET; (c) AOD; (d) SSA; (e) Ångström exponent.

[Figure]

Figure 12: Temporal variation of (a) AOD, (b) SSA, and (c) Ångström exponent from the SKYNET skyradiometer and AERONET sunphotometer under (a1, b1, c1) clean, (a2, b2, c2) light pollution, and (a3, b3, c3) heavy pollution weather conditions in Beijing on 27 December 2016, 2 January, and 4

Line 301: negation correlation should be negative correlation
**Response:** We have replace 'negation' with 'negative' in the revised manuscript. (Line 339)
'The correlation coefficient between visibility and RH is about − 0.80, showing a significant negative correlation during the study period.'

Section 4. The discussion section looks redudant. It should be included in the results or conslucions.
**Response:** The discussion section has removed, and it has been included in the results and consluctions.